# Fatty Acid Induced Hypermethylation in the *Slc2a4* Gene in Visceral Adipose Tissue Is Associated to Insulin-Resistance and Obesity

**DOI:** 10.3390/ijms24076417

**Published:** 2023-03-29

**Authors:** Jan H. Britsemmer, Christin Krause, Natalie Taege, Cathleen Geißler, Nuria Lopez-Alcantara, Linda Schmidtke, Alison-Michelle Naujack, Jonas Wagner, Stefan Wolter, Oliver Mann, Henriette Kirchner

**Affiliations:** 1Division Epigenetics & Metabolism, Institute of Human Genetics, University of Lübeck, 23562 Lübeck, Germany; 2Center of Brain, Behavior and Metabolism (CBBM), University of Lübeck, 23562 Lübeck, Germany; 3German Center for Diabetes Research (DZD), 23562 Lübeck, Germany; 4Institute of Endocrinolgy and Diabetes, University of Lübeck, 23562 Lübeck, Germany; 5Department of General, Visceral and Thoracic Surgery, University Medical Center Hamburg-Eppendorf, 20251 Hamburg, Germany

**Keywords:** *Slc2a4*, DNA methylation, human visceral adipose tissue, insulin resistance, diet-induced obese mice

## Abstract

De novo lipogenesis (DNL) in visceral adipose tissue (VAT) is associated with systemic insulin sensitivity. DNL in VAT is regulated through ChREBP activity and glucose uptake through Glut4 (encoded by *Slc2a4*). *Slc2a4* expression, ChREBP activity, and DNL are decreased in obesity, the underlying cause however remains unidentified. We hypothesize that increased DNA methylation in an enhancer region of *Slc2a4* decreases *Slc2a4* expression in obesity and insulin resistance. We found that *SLC2A4* expression in VAT of morbidly obese subjects with high HbA1c (>6.5%, *n* = 35) is decreased, whereas DNA methylation is concomitantly increased compared to morbidly obese subjects with low HbA1c (≤6.5%, *n* = 65). In diet-induced obese (DIO) mice, DNA methylation of *Slc2a4* persistently increases with the onset of obesity and insulin resistance, while gene expression progressively decreases. The regulatory impact of DNA methylation in the investigated enhancer region on SLC2A4 gene expression was validated with a reporter gene assay. Additionally, treatment of 3T3 pre-adipocytes with palmitate/oleate during differentiation decreased DNA methylation and increased *Slc2a4* expression. These findings highlight a potential regulation of *Slc2a4* by DNA methylation in VAT, which is induced by fatty acids and may play a role in the progression of obesity and insulin resistance in humans.

## 1. Introduction

Facing the global pandemics of obesity and type 2 diabetes (T2D), the discovery of new potential drug targets and preventive therapeutics is of high importance [1,2]. The solute carrier family 2 member 4 (*SLC2A4*) gene, encoding for the insulin-dependent glucose transporter type 4 (GLUT4) in muscle and adipose tissue, has therapeutic potential for the treatment of insulin resistance in obesity and in specific cancer types [3,4]. During basal conditions, GLUT4 protein is stored in intracellular vesicles. It is translocated into the cell membrane after an insulin stimulus to facilitate glucose uptake into the cell [5]. While the translocation of the GLUT4 protein in white adipose tissue (WAT) is impaired in insulin resistance, *SLC2A4* expression is already decreased in the proceeding obese state. This leads to reduced GLUT4 protein content and glucose uptake without affecting its translocation properties [6]. After glucose is transported into the adipocyte, the carbohydrate response element binding protein (ChREBP), encoded by the MLX-interacting protein-like (*MLXIPL*) gene, becomes activated and induces genes involved in de novo lipogenesis (DNL), for example, the key enzyme fatty acid synthase (*FASN*) [7,8]. Adipose tissue-derived DNL-lipokines, such as the fatty acid C16:1n7-palmitoelate, play a crucial role in regulating systemic metabolism and insulin sensitivity [9]. Therefore, *SLC2A4* gene expression is linked to DNL capacity in adipose tissue via ChREBP activity. The impact of *SLC2A4* expression on systemic insulin sensitivity and inter-organ crosstalk was studied in muscle-specific *Slc2a4* knock out (*Slc2a4*^−/−^) mice. *Slc2a4*^−/−^ mice display reduced muscle glucose uptake and increased insulin resistance [10]. This metabolic phenotype is reversed by adipose tissue-specific *Slc2a4* overexpression [10]. 

Currently, the mechanisms causing the observed downregulation of the *SLC2A4* gene in obesity are not known. In adipose tissue, the *Slc2a4* gene is regulated by several transcription factors, among them the specificity protein 1 (SP1) and the sterol regulatory element-binding transcription factor 1 (SREBP-1) [11,12]. Epigenetics are altered by lifestyle and environmental factors [13,14] and the association between epigenetic mechanisms, such as DNA methylation at CpG sites in the genome, and obesity and T2D is well established. Nevertheless, only a few data on the epigenetic regulation of *Slc2a4* in adipose tissue exist. In the visceral adipose tissue (VAT) of obese and insulin-resistant subjects, DNA methylation of the *SLC2A4* promoter is increased [15,16]. Accordingly, we hypothesized that DNA methylation of the *SLC2A4* in the proximity of an SP1 and SREBP-1 binding motif is increased in the VAT of obese humans. This might lead to impaired transcription factor binding and reduced *SLC2A4* gene expression. Indeed, we found that DNA methylation is dynamically altered and associated with reciprocal *Slc2a4* expression and fatty acids in obese humans and mice with insulin resistance. 

Herewith, we describe a novel epigenetic mechanism regulating the *SLC2A4* gene by fatty-acid-induced alterations in DNA methylation, which may play a role in the development of insulin resistance in obesity.

## 2. Results

### 2.1. Enhancer-Associated SLC2A4 DNA Methylation Is Increased in Visceral Adipose Tissue of Morbidly Obese Subjects with High HbA1c and Is Associated with Decreased Gene Expression

We hypothesized that an obesogenic diet, i.e., high fat content, induces DNA methylation of the glucose transporter 4 gene *SLC2A4* in VAT of obese subjects and that this epigenetic dysregulation leads to insulin resistance by reduced *SLC2A4* gene expression. Reduced *SLC2A4* gene expression putatively decreases glucose uptake into the adipocytes and in turn reduces ChREBP expression (*MLXIPL* gene), which subsequently results in decreased expression of DNL-associated genes (*FASN*, acetyl-CoA carboxylase 1 *(ACACA* alias *ACC1*)) (Figure 1a).

As our human cohort of morbidly obese patients stratified into subjects with low (≤6.5%) and high HbA1c (>6.5%) is lacking a non-obese control group, we started with analyzing publicly available transcriptomic data of subcutaneous adipose tissue (SAT) from non-obese (*n* = 26) and obese (*n* = 30) subjects (GSE25401, Figure 1b upper panel, Table 1) to validate the downregulation of *SLC2A4* expression in the adipose tissue of obese subjects. *SLC2A4* and *FASN* expression is significantly downregulated in SAT of obese versus non-obese subjects (Figure 1c and Appendix A). *MLXIPL* and *ACACA* gene expression is non-significantly reduced compared to the non-obese group (Figure 1c and Appendix A). *CD36*, which encodes for a fatty acid transporter, remains unchanged (Figure 1c and Appendix A), while pyruvate kinase M1/2 (*PKM*) gene expression is significantly increased in obese versus non-obese subjects (Figure 1c and Appendix A). We subsequently analyzed publicly available transcriptomic data from VAT of individuals with obesity (GSE20950) who are insulin sensitive (IS) or insulin resistant (IR) (Figure 1b lower panel), to test if there is a further decline of *SLC2A4* expression in the combined obese and insulin resistant state. Only expression of *ACACA* and *FASN* are significantly decreased in obese subjects with insulin resistance (Figure 1d and Appendix A). *SLC2A4*, *MLXIPL,* and *PKM* are slightly downregulated, whereas the fatty acid transporter, cluster of differentiation 36 (*CD36*), gene is non-significantly downregulated (Figure 1d and Appendix A). Importantly, this publicly available cohort is limited by a small number of study subjects with a relatively healthy metabolic profile (see Table 2). 

Therefore, we analyzed *SLC2A4* gene expression in the VAT of our human cohort, which has a larger sample size and more distinct metabolic differences between the two study groups. To find a molecular mechanism that could be causative for decreased *SLC2A4* expression, we studied DNA methylation in an enhancer region (GeneCards Enhancer ID: GH17J007279) that contains SP1 and SREBP-1c binding motifs. SP1 binding to the DNA is DNA methylation-sensitive [19]. We focused on the chromosomal area chr17:7282504-7282531 (ENST00000317370.13, UCSC) containing four CpG sites (Figure 2a, region of interest (ROI), reference genome hg38), which is located in intron 1 of *SLC2A4* and in the shore of a CpG island. SP1 binding in this region was previously confirmed by ChIP-Sequencing in HepG2 cells [20]. We stratified our cohort consisting of 101 individuals with a BMI > 35 into *low HbA1c* (≤6.5%, *n* = 65) and *high HbA1c* (>6.5%, *n* = 36) (Figure 2b). *SLC2A4* gene expression significantly decreased to 0.539-fold (±0.1) in obese subjects with high HbA1c compared to obese subjects with low HbA1c. *FASN* and *MLXIPL* gene expression is similarly decreased (Figure 2c) in the high HbA1c group, whereas *ACACA* gene expression is non-significantly decreased. Furthermore, *FASN* and *MLXIPL* expression correlate positively with *SLC2A4* gene expression (Figure 2d, *SLC2A4/FASN*: R = 0.3958, *p* = 0.0004; *SLC2A4/MLXIPL:* R = 0.3261). Though the group-wise difference of *ACACA* expression was not significant, we observed a weak significant positive correlation due to the reduced scattering (*p* = 0.0017; *SLC2A4/ACACA*: R = 0.3818, *p* = 0.0002). This is associated with increased DNA methylation at all four CpGs in the subjects of the high HbA1c group. DNA methylation at CpG1 increased by 2.225%, CpG2 by 1.965%, Cpg3 by 2.975%, and CpG4 by 4.02% absolute difference in DNA methylation (Figure 2e). To find potential factors that influence DNA methylation and *SLC2A4* gene expression, we performed correlation analysis with blood parameters, BMI, and age (Figure 2f, red box). *SLC2A4* gene expression correlates negatively with age (r = −0.263, *p* = 0.01, n = 101), HbA1c (r = −0.221, *p* = 0.032, n = 94), fasting glucose (r = −0.215, *p* = 0.04, n = 90), fasting insulin (r = −0268, *p* = 0.01, n = 87), HOMA index (r = −0251, *p* = 0.02, n = 83), serum C-peptide levels (r = −0219, *p* = 0.039, n = 89), and serum triglyceride levels (r = −0.248, *p* = 0.02, n = 94) (Figure 2f). Although *SLC2A4* DNA methylation does not directly correlate with *SLC2A4* gene expression in the entire cohort, we identified putatively relevant correlations in sub-cohorts that were stratified by medical indication to analyze possible effects of the medication on DNA methylation and gene expression. CpG4 DNA methylation negatively correlates with *SLC2A4* gene expression (Appendix A, r = −0.595, *p* = 0.041) and insulin levels (Appendix A, CpG1: r = −0.649, *p* = 0.042; CpG3: r = −0.689, *p* = 0.027; CpG4: r = −0.732, *p* = 0.016) in obese subjects with metformin treatment. Moreover, DNA methylation negatively correlates with glucose levels in individuals receiving insulin (Appendix A, CpG1: r = −0.719, *p* = 0.013; CpG2: r = −0.633, *p* = 0.036). Additionally, we performed a stepwise regression model. Inclusion of *SLC2A4* DNA methylation as a predictor into the regression increases the accuracy of modelling *SLC2A4* gene expression (R^2^ = 0.347; Adjusted R^2^ = 0.211; *p* = 0.00324; Figure 2g, Appendix A) compared to the stepwise-regression model excluding DNA methylation (R^2^ = 0.0691; Adjusted R^2^ = 0.059, *p* = 0.0105, Appendix A). 

In conclusion, *SLC2A4* gene expression is reduced in the VAT of obese individuals with high HbA1c and is accompanied by DNA hypermethylation at an SP1/SREBF binding site. Hence, *SLC2A4* gene expression correlates with serum triglyceride levels and *SLC2A4* might be regulated by fatty-acid-induced DNA methylation. 

### 2.2. Hypermethylation of the Slc2a4 Gene in VAT of Diet-Induced Obese Mice

To study if DNA methylation of the *SLC2A4* gene and the associated mRNA downregulation of *SLC2A4* and other genes relevant for DNL can be induced by lifestyle factors such as a Westernized diet rich in fat, we analyzed DNA methylation in our ROI and gene expression in VAT of a longitudinal diet-induced obese (DIO) mouse cohort (Figure 3a). This model allows for a differentiation between a causal or consequential effect of *Slc2a4* DNA methylation and DNL. The equivalent region of the human genome in the mouse genome was identified by a LiftOver analysis using the UCSC genome browser. Insulin resistance assessed by the Homeostasis Model Assessment of Insulin Resistance (HOMA-IR) is significantly increased in mice fed with HFD at 8 weeks (Chow: 0.50 ± 0.10; HFD: 3.17 ± 0.58) and 12 (Chow: 0.32 ± 0.06; HFD: 3.85 ± 1.70, Figure 3b).

To exclude that reduced *Mlxipl* gene expression and the resulting reduced DNL is attributed to the downregulation of glucose transporters other than *Slc2a4*, we measured *Slc2a1* gene expression, the gene encoding the insulin-independent glucose transporter Glut1. After an acute increase of *Slc2a1* gene expression to 2.31 (±0.47) fold in the HFD-fed mice at week 1, gene expression normalizes to the chow level over the time of HFD feeding, except for a short decrease at week 8 to 0.43 (±0.08) fold (Figure 3c). In contrast to the mainly unaltered *Slc2a1* gene expression, *Slc2a4* gene expression decreases over the investigated period. Gene expression is already significantly reduced to 0.55 (±0.06, *p* < 0.05) fold at week 4 and further decreases to 0.14 (±0.03, *p* < 0.0001) fold at week 12 (Figure 3d). Genes involved in glucose signaling and DNL show a similar pattern to *Slc2a4* expression, except for *Pkm* expression (Appendix A). After an acute increase at week 1, *Acc1*, *Fasn* (DNL), and *Mlxipl* (glucose signaling) gene expressions are significantly decreased at week 4 in HFD mice ((*Acc1*: 0.29 (±0.06; *p* < 0.0001, Figure 3e), *Fasn*: 0.42 (±0.11; *p* < 0.05, Figure 3f), *Mlxipl*: 0.44 (±0.06; *p* < 0.001, Figure 3g)). Gene expression gradually decreases over the time period in HFD-fed mice with its lowest point at week 12 ((*Acc1*: 0.09 (±0.01; *p* < 0.0001, Figure 3e), *Fasn*: 0.13 (±0.03; *p* < 0.0001, Figure 3f), *Mlxipl*: 0.17 (±0.04; *p* < 0.001, Figure 3g)). Glut4 protein levels tend to be reduced in the VAT of HFD mice without reaching statistical significance, likely due to fat contamination of the protein lysate which influenced the protein quantification and led to unequal amounts of protein being loaded to the gel (indicated by varying Hsp90 band intensities, Appendix A). Since DNA methylation of *SLC2A4* is increased in insulin-resistant humans, we analyzed *Slc2a4* DNA methylation in the VAT of mice. This region is highly conserved between the human and murine genomes, including the Sp1 binding motif (Figure 3h, reference genome = mm39). Both regions only differ in their number of CpG sites. The murine ROI exhibits two CpG sites at position chr11:69838231 (CpG1) and at position chr11:69838257 (CpG2). DNA methylation at both CpG sites gradually increases during HFD-feeding in the VAT of HFD mice. DNA methylation at CpG1 is significantly increased at week 1 prior to the decrease in gene expression (∆CpG1 = 4.004%, *p* = 0.0124, Figure 3i). The difference in DNA methylation between HFD mice and chow mice gradually increases to a final difference of ∆CpG1 = 18.01% (*p* < 0.0001) at week 12 (Figure 3i). DNA methylation at CpG2 was significantly increased at week 2 (∆CpG2 = 5.36%, *p* < 0.01) and gradually increases to a final difference of ∆CpG2 = 14.38% (*p* < 0.0001) at week 12 (Figure 3j). DNA methylation at both positions show a strong negative correlation with *Slc2a4* gene expression (CpG1: r = −0.5454, *p* < 0.0001; CpG2: r = −0.4942, *p* < 0.0001; Figure 3k). Furthermore, DNA methylation positively correlated with body weight, insulin, blood glucose, and hepatic triglyceride content (Figure 3k). That means that higher DNA methylation in the ROI of the *Slc2a4* is associated with obesity-related parameters of increased body weight, plasma insulin levels, and blood glucose levels. Additionally, *Slc2a4* gene expression positively correlates with the genes involved in DNL (*Fasn*, *Acc1*) and glucose signaling (*Mlxipl*). 

Overall, the findings observed in humans are reproducible in our longitudinal DIO mouse model, where increased DNA methylation in the ROI is associated with a lower expression of the *Slc2a4* gene. Furthermore, a clear correlation between DNA methylation and gene expression exists, pointing at an epigenetic regulation of the *Slc2a4* gene that may be masked in the human cohort due to different medications or lifestyle factors. Furthermore, the mouse model shows that this epigenetic dysregulation establishes in a very early stage of obesity prior to the onset of insulin resistance. FFA might regulate *SLC2A4* gene expression via altering DNA methylation in the ROI since *Slc2a4* gene expression and DNA methylation correlate with hepatic triglyceride content in mice and with triglyceride plasma levels in humans.

### 2.3. Palmitate/Oleate Treatment of Differentiating Preadipocytes Alters DNA Methylation in the Slc2a4 Gene

To test the hypothesis that FFA may increase *Slc2a4* DNA methylation and therefore contribute to the downregulation in *Slc2a4* gene expression, we treated 3T3 preadipocytes during differentiation with palmitate/oleate (PO) over a period of 14 days after the induction of differentiation (Figure 4a). During PO treatment, *Slc2a4* gene expression increases to 1.85 (±0.23) fold at day 5 after the start of induction which we believe to be a compensatory reaction. With ongoing PO treatment and differentiation, *Slc2a4* gene expression remains increased relative to the BSA control, however, the difference between the BSA control and PO treatment diminishes with PO-treated cells being only 1.29 (±0.14) fold over the BSA control at D14 (Figure 4b). Importantly, from D10 on, the difference between PO and BSA treatment does not reach statistical significance anymore. The onset of a decrease in Slc2a4 expression at day 10 relative to D5 is accompanied by a relative decrease of the Slc2a4-downstream genes *Mlxipl* and *Fasn*, but not *Acc1* (Figure 4c–e). It should be noted that the absolute expression of *Slc2a4* (normalized to BSA control of D5) increases in both treatment groups over the course of differentiation (Appendix A). This matches with the decrease in Slc2a4 DNA methylation observed, particularly at CpG2 (Figure 3g).

These findings are coherent with the observations in the VAT of HFD mice, where DNA methylation and *Slc2a4* gene expression correlate inversely. The highest difference in DNA methylation is observed at day 10 in CpG2 (∆CpG2 = 8.4%) and at day 14 in CpG1 (∆CpG1 = 5.17%, Figure 3f,g). With ongoing treatment, *Slc2a4* gene expression and DNA methylation normalize again, with no significant differences compared to the BSA control (Figure 4b,f,g). To additionally validate that methylation in our ROI in the proximity of an Sp1 and SREBPF-1 binding motif influences gene expression, we performed a methylation-sensitive reporter gene assay (Figure 4h). The ROI was cloned into a CpG-free luciferase vector and the luciferase signal was measured in the unmethylated (0% methylated) and completely methylated state (100% methylated). In the fully methylated state, the luciferase signal decreases to 77.53% (±16.71%, *p* = 0.039). This result aligns with the previous findings in mice, humans, and differentiating 3T3 preadipocytes, where higher levels of DNA methylation are associated with lower *Slc2a4* gene expression.

## 3. Discussion

Expression of the glucose transporter GLUT4 is reduced in obesity but the underlying mechanism is incompletely understood. We hypothesized that decreased *SLC2A4* gene expression in the VAT of obese humans with impaired glycemic control (high HbA1c) emerges from a fatty-acid-induced (dietary or obesity-associated) increase in DNA methylation in a *SLC2A4* enhancer region spanning several SP1 and SREBPF-1 binding motifs. The downregulation of *SLC2A4* may consequently be linked to the reduced DNL capacity of adipose tissue in obesity [7,8,21], which interferes with the inter-organ crosstalk to regulate systemic insulin sensitivity through adipose-tissue-derived lipokines [9] and promotes the development of insulin resistance in obese individuals.

We confirmed our hypothesis that high concentrations of fatty acids dynamically modify *Slc2a4* DNA methylation and expression in adipocytes, which could explain the increased DNA methylation seen in obese humans and DIO mice that have increased triglyceride levels in their circulation or liver and are exposed to a diet rich in fat. Using publicly available transcriptomic data of non-obese and obese subjects, we confirmed the decreased *SLC2A4* gene expression in the SAT of obese subjects (GSE25401). However, unlike in our own cohort, no significant downregulation of *SLC2A4* gene expression in the VAT of obese insulin-resistant subjects compared to obese insulin-sensitive was found in dataset GSE20950. This discrepancy between our own and public data might be explained by differences in serum triglyceride levels compared to our high HbA1c subjects, among others. Only in our cohort were serum triglycerides increased in the insulin-resistant group which likely triggered *SLC2A4* DNA methylation. However, further differences between our and the public cohort existed (indicated by increased age, basal glucose levels, fasting insulin levels, and HOMA-IR values) compared to the IR subjects from GSE20950 (Table 2 and Table 3). This assumption is supported by Roux-en-Y gastric bypass (RYGB) studies in obese subjects. At 24 weeks post-surgery, *SLC2A4* gene expression in adipose tissue is restored, which is associated with an improved metabolic profile (reduced HbA1c, reduced fasting insulin levels) [22].

The observed decreased *SLC2A4* gene expression in VAT of obese subjects was accompanied by a site-specific increase in DNA methylation in the proximity of an SP1/SREBPF-1c binding motif. Since DNA methylation impairs protein/DNA interactions, the increased DNA methylation likely leads to decreased SP1/SREBPF-1c binding [21,23]. This results in decreased gene expression, as SP1 and SREBPF-1c are known enhancing transcription factors that promote *SLC2A4* gene expression [12]. We already reported in the same human cohort that increased DNA methylation in the proximity of an SP1 and SREBPF-1c binding motif of the *FASN* gene, encoding the key regulator enzyme of DNL, is associated with its downregulation [21]. This supports the hypothesis that DNA methylation plays a crucial role in regulating genes associated with glucose and lipid homeostasis in adipose tissue. Surprisingly, *SLC2A4* gene expression did not correlate directly with DNA methylation in our human cohort. Nevertheless, including DNA methylation of the *SLC2A4* gene as a cofactor in a stepwise regression model improved the accuracy to predict *SLC2A4* gene expression in comparison to excluding DNA methylation. This demonstrates the regulatory role of DNA methylation on *SLC2A4* gene expression. As it is known that anti-diabetic medication such as metformin can alter DNA methylation [24], we correlated DNA methylation with *SLC2A4* gene expression of sub-groups of obese individuals treated with metformin (Obese Met) or insulin (Obese Ins). DNA methylation negatively correlated with *SLC2A4* expression in obese subjects receiving metformin but not insulin. This indicates that insulin treatment may alter *SLC2A4* gene expression without affecting DNA methylation. In this line, it has already been shown that insulin induces *SLC2A4* gene expression independently of SP1 and SREBF-1c [25]. However, these data should be interpreted with caution as the sample size was very low.

*SLC2A4* gene expression correlated with serum triglyceride levels. This emphasizes the important role of fatty acids to regulate the glucose homeostasis of adipose tissue in obesity. Similar effects of fatty acids on *Slc2a4* gene expression in adipose tissue could be observed in rats, where treatment with atorvastatin, a drug applied for hypercholesterolemia, restored *Slc2a4* gene expression by lowering serum cholesterol and triglycerides [26,27]. Our findings were reproducible in a DIO mouse model. Importantly, these changes already occur at a very early stage after HFD feeding. DNA methylation was significantly increased at week 2 before *Slc2a4* gene expression decreased, which supports our hypothesis that DNA methylation at this region is rather a causative mechanism in regulating *Slc2a4* in obesity than a consequential response to high-fat feeding. Our assumption that DNA methylation in the ROI is sensitive to fatty acids is reinforced by the treatment of 3T3 preadipocytes with palmitate and oleate during differentiation. Apart from the fact that DNA methylation is changing during differentiation [28], this process seems to be accelerated by PO treatment in 3T3 preadipocytes. Methylation in the ROI starts to decrease in the first days after PO treatment while absolute *Slc2a4* gene expression increases (Appendix A). This is counterintuitive to the expectations based on the observations made in vivo, where gene expression decreases and methylation increases during the period of high-fat feeding. This acute induction phase of *Slc2a4* gene expression also occurred in the DIO mouse model during the first week of high-fat feeding and seems to be regulated by CpG2. Chronic conditions of high fatty acid levels on the other hand, which frequently occur in obesity and are termed lipotoxicity, lead to a decrease in *Slc2a4* gene expression in the mouse and cell culture models. Moreover, the functional role of the investigated CpG sites on gene expression could be mechanistically validated by a methylation-sensitive reporter gene assay. However, the use of 3T3-derived adipocytes is a limitation. The effect of FA on *Slc2a4*-DNA methylation has still to be elucidated in primary adipocytes or directly in obese subjects. Additionally, we cannot determine the exact mechanism of how DNA methylation is dynamically changed by fatty acids. Collectively, our findings provide a new epigenetic pathomechanism in the regulation of the *Slc2a4* gene in obesity in which dietary or endogenously derived FA may increase enhancer-associated DNA methylation in the *Slc2a4* gene to partially cause a downregulation of *Slc2a4* expression which ultimately reduces glucose uptake and contributes to insulin resistance. Our findings may also have implications for dietary and pharmacological therapies in diabetes, especially regarding diet composition and blood lipids. Often, the emphasis is on the reduction of carbohydrate intake since the primary symptom in T2D is dysregulated glucose homeostasis [29]. In the future, the physiological impact of DNA methylation in the ROI on the organism needs to be elucidated by, for example, modifying specifically these CpGs in mice, to protect them from the development of insulin resistance.

## 4. Materials and Methods

### 4.1. Analysis of Publicly Available Transcriptomic Data

Publicly available transcriptome data generated from human subcutaneous adipose tissue or visceral adipose tissue with accession numbers GSE25401 [30] and GSE20950 [31] were downloaded from the NCBI gene expression omnibus. The GSE25401 cohort comprises 56 individuals in total, which were previously separated into obese (*n* = 30; BMI > 35) and non-obese (*n* = 26; BMI < 27) individuals (Table 1). The GSE20950 cohort comprises 20 individuals in total, which were previously separated into insulin-sensitive (IS; *n* = 10; HOMA < 2.4) and insulin-resistant (IR; *n* = 10; HOMA ≥ 2.4) by HOMA-IR (Table 2). SP1 binding motif was identified using the Berg and Hippel weighted algorithm and JASPAR binding motif MA0079.2.

### 4.2. Study Subjects Used for qPCR and Pyrosequencing Analysis

Visceral adipose tissue biopsies were taken from 101 obese subjects (Table 3) during bariatric surgery at the University medical center Hamburg-Eppendorf and flash frozen until further analysis. All participants signed informed consent and the study was approved by the local ethics committee (PV4889). Subjects were grouped by their HbA1c blood level into low HbA1c (L-HbA1c; *n* = 65; HbA1c ≤ 6.5%) and high HbA1c (H-HbA1c; *n* = 36; HbA1c > 6.5%) subgroups. Blood parameters were measured by the Institute of Clinical Chemistry and Laboratory Medicine, University Medical Center Hamburg-Eppendorf. Not all clinical parameters were measured in the entire cohort. 

In total, 11 individuals received insulin treatment and 15 metformin and additional oral antidiabetics (not known). DNA methylation in VAT of other genes was previously studied by us in a subset of this cohort [21].

### 4.3. Mouse Model of Diet-Induced Obesity

Male C57BL/6N mice were obtained from Charles River at 4 weeks of age and were housed under constant temperature (22 ± 1 °C) with a normal 12-h light/dark cycle. Mice had ad libitum access to water and food. After one week of acclimation, mice were randomized into two groups of similar body weights. Mice were fed consistently ad libitum with either a chow diet (=control group; Breeding Diet 1314 obtained from Altromin, Germany) or a 60% high-fat diet (=treated group; HFD, D12492, Research Diets, New Brunswick, NJ, USA) for 1, 2, 4, 5, 6, 7, 8 or 12 weeks. At each indicated time point, an intraperitoneal glucose tolerance test was performed (data shown in [32] and Appendix A). On the day of sacrifice, mice were anesthetized with ketamine/xylazine (120 mg/kg ketamine and 16 mg/kg xylazine) without recovery and blood was obtained by cardiac puncture. Plasma was prepared from EDTA-blood by centrifugation for 15 min at 2000× *g* and stored at −80 °C. Subsequently, mice were perfused with Krebs-Ringer solution with 1 U/mL heparin, the VAT was removed and immediately frozen and stored at −80 °C. The experiments involving the handling of mice were non-blinded. All procedures were in accordance with the local animal welfare guidelines and the EU Directive 2010/63/EU on the protection of animals used for scientific purposes approved. All procedures were approved by the MELUR Schleswig–Holstein, Germany (V 242-59721/2016). For further information regarding the data of plasma insulin, blood glucose, hepatic triglycerides levels, and bodyweight, see Geißler et al. [32] and Appendix A. Homeostasis Model Assessment of insulin resistance (HOMA-IR) was calculated as described by Fraulob et al. [30]:(1)$$HOMA-IR=\frac{{glucose}_{basal}\left[\frac{mmol}{L}\right]\times {insulin}_{basal}[\frac{IU}{mL}]}{22.5}.$$

### 4.4. RNA Extraction, cDNA Synthesis and RT-qPCR

RNA was extracted with the miRNeasy^®^ Mini Kit (Qiagen, Germany) using the protocol for RNA purification from tissues including on-column DNase. A total of 1000 or 2000 ng RNA of each sample was reverse transcribed into cDNA using SuperScriptTM IV VILOTM (ThermoFisherScientific, Waltham, MA, USA for human samples) or using the High-Capacity cDNA Reverse Transcription Kit (ThermoFisherScientific U.S., for mouse samples) according to manufacturer’s instructions. For gene expression quantification from human VAT, Real-time PCR was performed in a fast thermal cycling protocol (TagMan Fast Advanced Master Mix (ThermoFisher Scientific, U.S.)) using 12.5 ng cDNA in a total volume of 10 µL per well. Normfinder analysis was performed to find the best housekeeping gene *RPL37A* (Hs01102345_m1, ThermoFisher Scientific, U.S.). Gene expression levels of *SLC2A4* (Hs00168966_m1), *FASN* (Hs01005622_m1), *ACACA* (Hs01046047_m1), and *MLXIPL* (Hs00975714_m1) were measured with TaqMan assays (ThermoFisher Scientific, U.S.) by real-time PCR in duplicates. For gene expression quantification from murine VAT, Real-time PCR was performed in a standard thermal cycling protocol (PrimeTime™ Gene Expression Master Mix, IDT) using 10 ng cDNA in a total volume of 10 µL per well. Normfinder analysis was performed to find the best housekeeping gene *Hprt1* (Mm.PT.39a.22214828, IDT). Gene expression levels of *Slc2a4* (Mm.PT.58.9683859), *Fasn* (Mm.PT.58.14276063), *Acc1* (Mm.PT.58.12492865), *Mlxipl* (Mm.PT.56a.33592172), *Pkm* (Mm.PT.58.6642152), and *Slc2a1* (Mm.PT.58.7590689) were measured with PrimeTime assays (IDT) in duplicates. Data were analysed with the Pfaffl method [31]. For correlations with DNA methylation in mice and metabolic parameters, the −dCt (Ct(housekeeper) − Ct(target)) was calculated.

### 4.5. DNA Extraction from VAT, Bisulfite Conversion, and Pyrosequencing

DNA was extracted with the QIAamp^®^ DNA Mini Kit (Qiagen, Germany) following the protocol for DNA purification from tissues. A total of 560 ng (human) or 1000 ng (murine) DNA of each sample were treated with sodium-bisulfite with the EpiTect Fast DNA Bisulfite Kit (Qiagen, Germany) using the protocol for low-concentration samples or normal concentration samples. Bisulfite-PCR was carried out with the PyroMark PCR Kit (Qiagen, Germany) using 14 ng (human) or 20 ng (murine) bsDNA per reaction in a total volume of 25 µL. Bisulifte-PCR-Primers are described in Appendix A. Primers were designed with the PyroMark Assay Design 2.0 Software V2.0.1.15 (Qiagen, Germany). Pyrosequencing was performed on a Pyromark Q48 Autoprep with PyroMark Q48 Advanced CpG Reagents (Qiagen, Germany) using 10 µL of the PCR product. Pyromark Q48 Autoprep V2.4.2 software (Qiagen, Germany) was used for data analysis. Bisulfite-treated control DNA with known methylation quantities (EpiTect control DNA, QIAGEN, Hilden, Germany) was used for assay establishment and validation. Systematic quality controls were performed before pyrosequencing, including tests for primer-dimers and primer self-annealing. These quality controls included sequencing of (i) reverse and sequencing primer w/o PCR product, (ii) reverse primer only, (iii) sequencing primer only, and (iiii) PCR product w/o primers.

### 4.6. Western Blot

A total of 50 mg of VAT tissue from mice were homogenized in RIPA buffer with proteinase inhibitors (cOmpletete^TM^ protese inhibitor cocktail, Roche, Basel, Switzerland). Protein concentrations were measured using the Qubit Protein BR Assay (Qiagen, Germany). A total of 40 µg of total protein were diluted in loading buffer (Blue protein loading dye, New England BioLabs, Ipswich, MA, USA) according to manufacturer’s instructions and loaded onto 10% Criterion^TM^ Stain-Free^TM^ Protein Gels (Bio-Rad, Hercules, CA, USA). Proteins were transferred onto a Trans-Blot Turbo Midi 0.2 µm PVDF membrane (Bio-Rad, USA) using the Trans-Blot Turbo Transfer System (Bio-Rad, USA). PVDF membrane was blocked in 5% milk powder/TBS for 1 h at room temperature. Then, the membrane was incubated with anti-Glut4 antibody (1:1000 in 5% milk powder/TBS; Proteintech: 66846-1-Ig, USA) overnight at 4 °C. The membrane was washed three times with TBS-T for 20 min each and then incubated with secondary antibody-horseradish peroxidase (HRP) conjugate (Agilent, Santa Clara, CA, USA, P0447). The HRP signal was measured with Clarity Max Western ECL Substrate (Bio-Rad, USA) according to manufacturer instructions. The membrane was washed with TBST for 5 min at room temperature and then stripped using 5 mL RestorePLUS Western Blot Stripping buffer (ThermoFisher, USA) for 10 min at room temperature. Afterwards, the membrane was washed three times in TBST for 15 min each. Membrane was blocked again using 5% milk powder/TBS, followed by an incubation with the anti-HSP90 antibody (Cell Signaling Technologies, Danvers, MA, USA, 4877S). The washing steps, incubation with secondary antibody, and measurement of HRP-signals were performed in the same way as described for the anti-Glut4 antibody. The HRP-signals were analyzed using the Image Lab software 6.0.1 from Bio-Rad. 

### 4.7. Methylation-Sensitive Reporter Gene Assay

The region of interest (ROI) chr11:69838231-69838257 (mm39) was amplified from genomic murine DNA using primers with restriction enzyme overhangs (sequence in Appendix A). The insert was purified from an agarose gel using the Wizard PCR Clean-Up Kit (Promega, Madison, WI, USA) and cloned into the pCpGfree-mcs_v2 vector (InvivoGen, San Diego, CA, USA) at the HindIII (New England BioLabs, USA) and NcoI (New England BioLabs, USA) restriction sites using the QuickLigation™ Kit (New England BioLabs, Ipswich, MA, USA). Correct insertion was checked by sequencing using the vector_seq primer (Appendix A). A total of 50 ng of the pCpGfree-mcs_v2-Slc2a4 insert vector was co-transfected with 1 ng of reference Renilla vector (pRL; Promega, USA) per well of a 96-well plate into HEK293T cells using 0.5 µL of Lipofecatmine 2000 (ThermoFisherScientific, USA) and incubated for 24 h at 37 °C and 5% CO_2_. For measuring firefly luciferase and renilla signal, the Dual-Glo^®^ Luciferase Assay System (Promega, USA) was used according to manufacturer’s instructions.

### 4.8. Treatment of 3T3 Preadipocytes with Palmitate/Oleate Mixture

3T3-L1 preadipocytes (ATCC, Manassas, VA, USA) were cultured in proliferation medium (gibco DMEM with 4.5 g/L glucose, 10% FBS, 1% Pen/Strep) at 37 °C and 5% CO_2_. For differentiation timeline and treatment with 0.5 mM palmitate/oleate mixture (0.25 mM palmitate + 0.25 mM oleate), 2 × 10^4^ 3T3 preadipocytes were seeded into one well of a 12-well plate (Sarstedt, Nümbrecht, Germany) and were cultured in 1 mL proliferation medium for four days at 37 °C and 5% CO_2_ until confluent. The seeding day was considered day-4. Once the cells reached confluency, the initiation of differentiation was started using initiation medium (proliferation medium, 0.25 µM dexamethasone, 0.5 mM 3-isobutyl-1-methylxanthin, 1 µg/mL insulin) together with a treatment of 0.5mM palmitate/oleate or 0.75% BSA as control, considered day 0. After 72 h of incubation (day 3), the medium was changed to differentiation medium (proliferation medium, 1 µg/mL insulin), together with a second treatment of 0.5 mM palmitate/oleate or 0.75% as control. Further treatments were performed at day 5, 7 and 9. At day 11, the palmitate/oleate treatment was stopped and medium was replaced by differentiation medium containing no palmitate/oleate or BSA. Cells were harvested at day 5, 7, 10 and 14. Palmitate (Sigma-Aldrich, St. Louis, MO, USA, P9767) and oleate (Sigma, O7501) were solved in 0.1 M NaOH at 70 °C and diluted in 7.5% BSA to final concentration of 5 mM. Cells were not immediately harvested after acute treatment with palmitate/oleate. 

### 4.9. Statistical Analysis

qPCR data are presented as mean fold-change ± SEM according to the Pfaffl method [31]. The dCt values were used for statistical analysis. For statistical analysis of *SLC2A4* gene expression and DNA methylation from human VAT was performed using Mann-Whitney U test, after testing for normal distribution with a D’Agostino & Pearson normality test. For statistical analysis of gene expression and DNA methylation from murine VAT, a Two-Way ANOVA was performed. For statistical analysis of the luciferase assay, an unpaired, two-sided *t* test was performed. Correlation matrices and volcano plots were created and calculated in Python using the packages pandas (V. 1.5.2), bioinfokit (V. 2.1.0), matplotlib (V. 3.5.2), seaborn (v. 0.11), numpy (V. 1.23.0), and SciPy (V. 1.9.1.) [33]. Correlations were adjusted for multiple testing using the Benjamin-Hochberg method. 

### 4.10. Stepwise Regression Model

The stepwise regression model was done in MatLab (2020a, MathWorks, Natick, MA, USA). The formula of the stepwise regression was generated as:
(2)*SLC2A4* gene expression = 1 × β_0_ + (OPMI × Age)β_1_ + (OPBMI × *SLC2A4*_CpG1)β_2_ + (OPBMI × SLC2A4_CpG2)β_3_ + (OPBMI × SLC2A4_CpG4)β_4_ + (Age × sex_2_w_1_m)β_5_ + (Age × SLC2A4_CpG1)β_6_ + (SLC2A4_CpG1 × SLC2A4_CpG2)β_7_ + (SLC2A4_CpG2 × SLC2A4_CpG3)β_8_ + (SLC2A4_CpG3 × SLC2A4_CpG4)β_9_

In which only the interaction terms remained significant. Corresponding estimates (β) and *p* values are listed in Appendix A for the final formula.

## Figures and Tables

**Figure 1 ijms-24-06417-f001:**
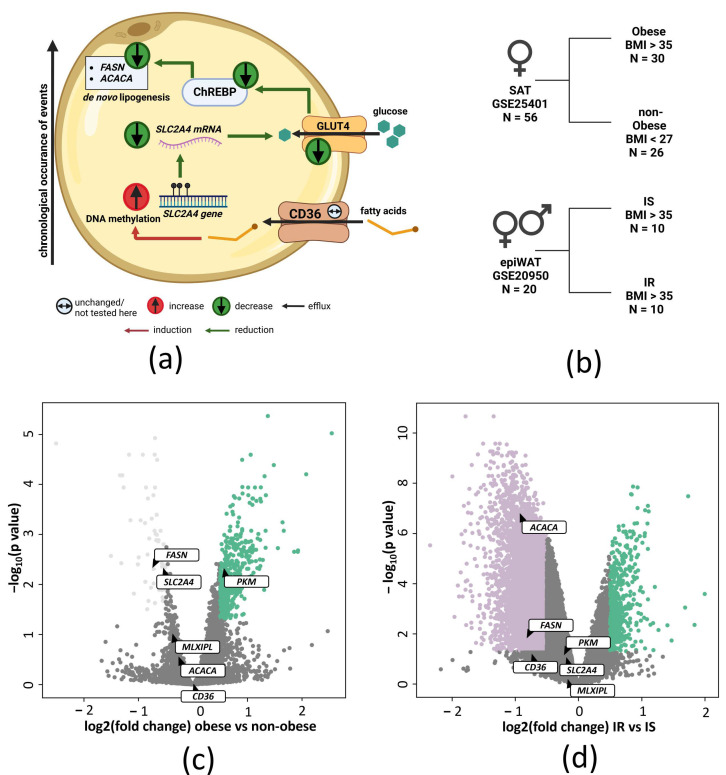
SLC2A4 and DNL-associated gene expression in publicly available transcriptomic data. (**a**) Schematic hypothesis. Fatty acid uptake into the adipocyte may alter DNA methylation of the *SLC2A4* gene, which leads to decreased GLUT4 protein levels and reduced insulin-stimulated glucose uptake. Consequently, ChREBP activity and de novo lipogenesis decrease. (**b**) Human cohorts of publicly available transcriptomic data used for analysis. (**c**) Volcano plot of publicly available transcriptome data (GSE25401) from subcutaneous adipose tissue of non-obese and obese normoglycemic individuals. Highlighted are genes involved in de novo lipogenesis (ACACA, FASN, PKM), glucose metabolism (SLC2A4, MLXIPL), and fatty acid transport (CD36). Colored dots represent genes that fulfill the criteria: *p*-value < 0.05, log2 Fold change > −0.5 or 0.5; (**d**) Volcano plot of publicly available transcriptome data (GSE20950) from visceral adipose tissue of obese, insulin sensitive (IS) and obese, insulin resistant (IR) subjects. Highlighted are the same genes as in (**c**). Colored dots represent genes that fulfill the criteria: *p*-value < 0.05, log2 Fold change > −0.5 or 0.5 epiWAT–epididymal white adipose tissue, SAT–subcutaneous adipose tissue, IS–insulin sensitive, IR–insulin resistant.

**Figure 2 ijms-24-06417-f002:**
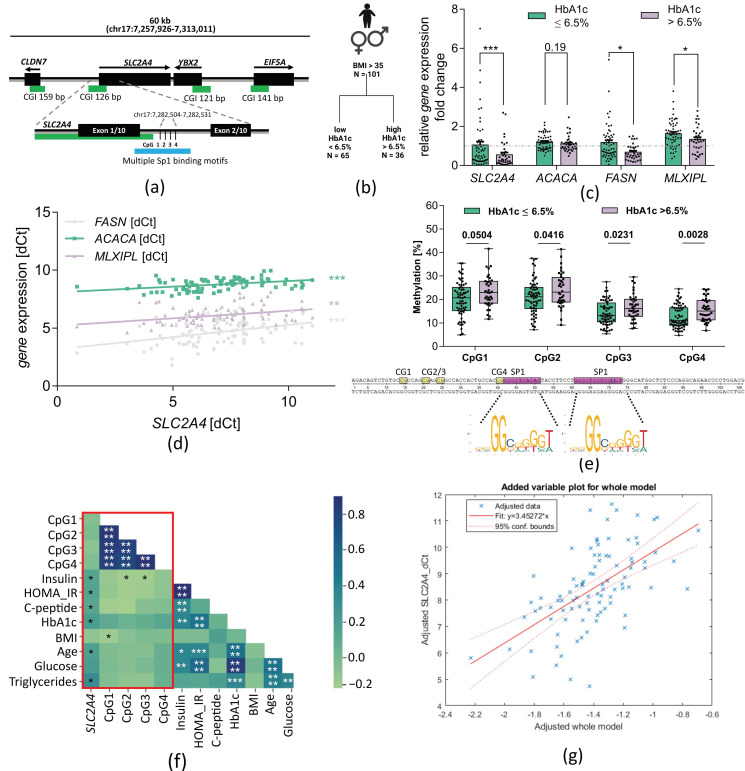
*SLC2A4* gene expression and DNA methylation in humans. (**a**) Overview of the SLC2A4 gene and the CpG sites that were analyzed for DNA methylation. The ROI is downstream of the first exon and includes four CpG sites in the proximity of multiple Sp1 binding motifs. Green bars indicate CpG islands, black bars indicate genes or exons; (**b**) Overview of the human cohort used for analysis of DNA methylation and gene expression in VAT; (**c**) Relative gene expression of SLC2A4 (l-HbA1c = 58; h-HbA1c = 35), ACACA (l-HbA1c = 57; h-HbA1c = 35), FASN (l-HbA1c = 58; h-HbA1c = 35), and MLXIPL (l-HbA1c = 56; h-HbA1c = 35) in VAT of morbidly obese individuals. Gene expression was measured with RT-qPCR. Fold change to obese individuals with low HbA1c is shown as mean ± SEM. Mann-Whitney test was performed after testing for normal distribution using the D’Agistino&Pearson normality tests. (**d**) Pearson correlation of *SLC2A4*, *ACACA*, *FASN,* and *MLXIPL* gene expression; (**e**) Methylation in ROI in VAT of morbidly obese subjects with low (n = 65) and high HbA1c (n = 36). Methylation was measured by bisulfite-pyrosequencing. Median ± Min./Max. is shown. Mann-Whitney test was performed after testing for normal distribution using D’Agistino&Pearson normality tests; (**f**) Correlation matrix of clinical parameters with *SLC2A4* gene expression and methylation in VAT. Blue squares indicate positive correlation coefficients and green squares negative ones. * = *p* < 0.05, ** = *p* < 0.01, *** = *p* < 0.001, **** = *p* < 0.0001. Pearson correlation was performed. Correlation analysis with *SLC2A4* gene expression was conducted using –dCt values; (**g**) Adjusted stepwise regression model for *SLC2A4* gene expression prediction.

**Figure 3 ijms-24-06417-f003:**
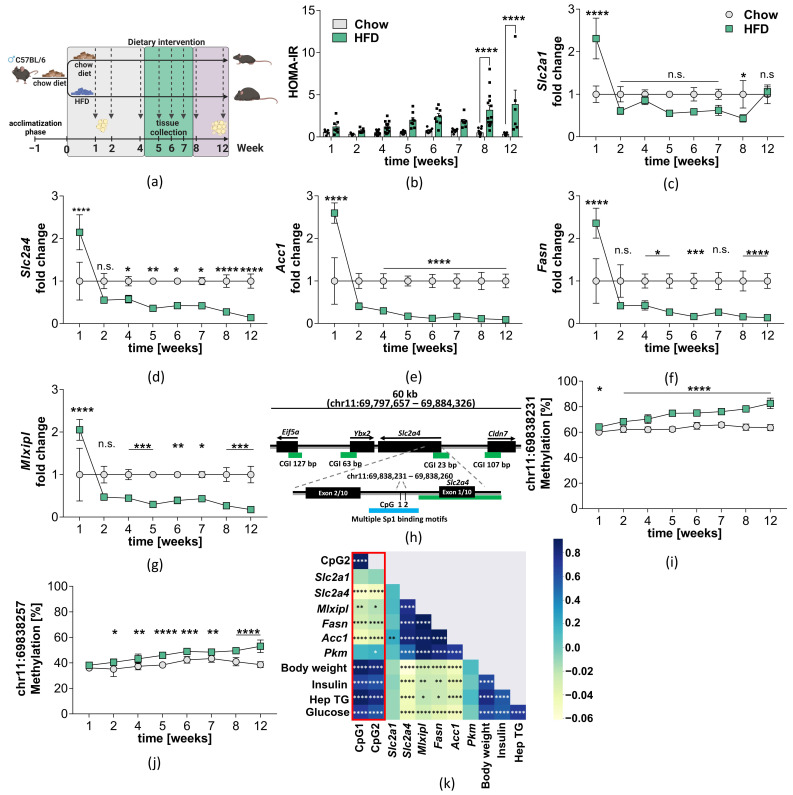
Gene expression and DNA methylation of the Slc2a4 gene in VAT of DIO mice fed a high-fat diet. (**a**) Feeding regime of high-fat mice over a period of 12 weeks. Time points of sacrifices are indicated in weeks. Colored background demonstrates stages of obesity and insulin resistance in HFD mice, with grey = lean; green = obese; purple = obese and insulin-resistant; (**b**) HOMA-IR for Chow (*n* = 5–14) and HFD (*n* = 6–14) fed mice. Mean ± SEM is shown; (**c**–**g**) Expression of genes involved in glucose uptake (*Slc2a1*, *Slc2a4*), glucose signalling (*Mlxipl*), and de novo lipogenesis (*Acc1*, *Fasn*) in VAT. Mean ± SEM of fold change between HFD (n = 7–16) and chow (n = 8–16) is shown, after normalizing expression to Hprt. A Two-Way ANOVA was performed with n.s. = non-significant, * = *p* < 0.05, ** = *p* < 0.01, *** = *p* < 0.001, **** = *p* < 0.0001; (**h**) Schematic overview of region of interest in mouse genome. The ROI is downstream of the first exon and includes two CpG sites in the proximity of multiple Sp1 binding motifs. Green bars indicate CpG islands and black bars genes/exons in this area; (**i**,**j**) Methylation in VAT at CpGs in the ROI. The chromosomal coordinates of the CpGs are chr11:69838231 (*CpG1*) and chr11:69838257 (*CpG2*). Mean ± SD of HFD (n = 7–16) and chow (n = 8–16) is shown. A Two-Way ANOVA using the Sidak correction for multiple testing was performed with n.s. = non-significant, * = *p* < 0.05, ** = *p* < 0.01, *** = *p* < 0.001, **** = *p* < 0.0001; (**k**) Pearson Correlation matrix of *Slc2a4* DNA methylation with metabolic parameters and gene expression. Blue indicates positive correlation and yellow negative. For correlation of gene expression, the -dCt value was used.

**Figure 4 ijms-24-06417-f004:**
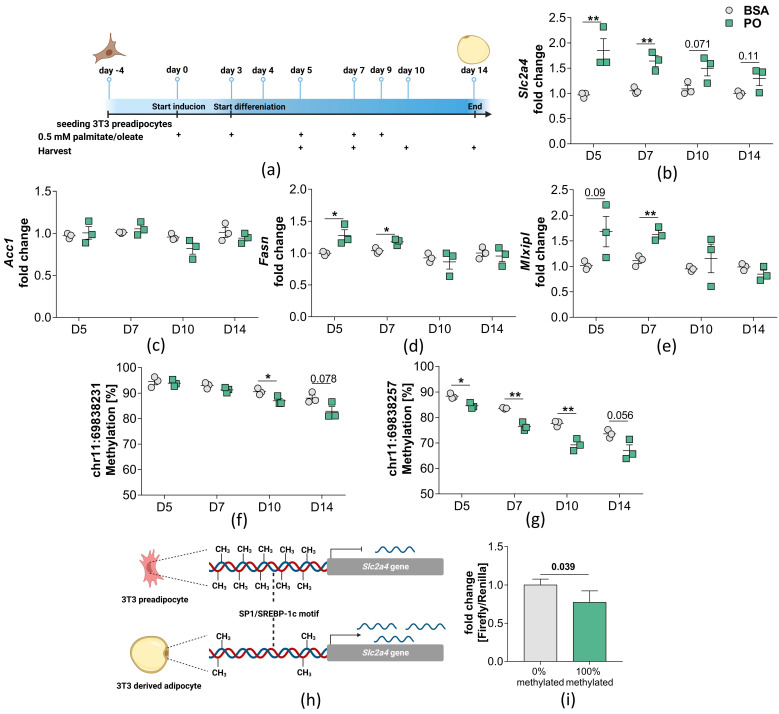
Gene expression and DNA methylation in 3T3 cells after treatment with palmitate/oleate. (**a**) Treatment regime of 3T3 cells during differentiation with either 0.5 mM palmitate/oleate (P/O, ratio 1:1) or 0.75% BSA (vehicle). 3T3 cells were seeded at D-4, start of induction of differentiation was performed at D0 and start of differentiation at D3. Crosses indicate time points of treatment. Cells were harvested at D4, D5, D7, and D10 for DNA and RNA extraction. (**b**–**e**) expression of genes involved in glucose uptake (*Slc2a1*, *Slc2a4*), glucose signalling (*Mlxipl*), and de novo lipogenesis (*Acc1*, *Fasn*). Mean ± SEM of fold change between P/O treatment (*n* = 3) and BSA treatment (*n* = 3) is shown, after normalizing expression to Hprt. A Two-Way ANOVA was performed with, * = *p* < 0.05, ** = *p* < 0.01 (**f**,**g**) DNA methylation in the ROI under palmitate/oleate treatment or vehicle. Mean ± SD A Two-Way ANOVA was performed with * = *p* < 0.05, ** = *p* < 0.01. (**h**) Scheme on how DNA methylation may impair transcription factor binding and influence gene expression; (**i**) Methylation sensitive reporter gene assay. ROI was amplified from genomic murine DNA and cloned into a CpG-free reporter gene vector. Afterwards, ROI was methylated using a DNA methyltransferase and the vector was transfected into HEK293T cells. Displayed is mean ± SD after normalizing to 0% methylated as control. For statistical analysis unpaired, two-sided students t-test was performed.

**Table 1 ijms-24-06417-t001:** Metabolic parameters of the GSE25401 cohort (modified [17]).

Characteristic	Obese (*n* = 30)	Non-Obese (*n* = 26)	*p* Value
Age [years]	43 ± 2	43 ± 3	0.8
BMI ^1^ [kg/m^2^]	40.9 ± 1.3	24 ± 1.8	<0.0001
Log_10_ HOMA_IR_ ^2^	0.458 ± 0.06	−0.01 ± 0.049	<0.0001

^1^ Body-mass-index (BMI); ^2^ homeostasis model assessment (HOMA).

**Table 2 ijms-24-06417-t002:** Metabolic parameters of the GSE20950 cohort (modified [18]).

Characteristic	Insulin Sensitive (HOMA < 2.4; *n* = 10)	Insulin Resistant (HOMA > 2.4; *n* = 10)	*p* Value
Age [years]	39 ± 6	43 ± 9	0.2 (n.s.)
Woman [number]	8	6	0.4 (n.s.)
BMI ^1^	48 ± 3	49 ± 7	0.7 (n.s.)
Total cholesterol [mg/dL]	176 ± 27	179 ± 50	0.9 (n.s.)
LDL cholesterol ^2^ [mg/dL]	109 ± 15	101 ± 45	0.6 (n.s.)
HDL cholesterol ^3^ [mg/dL]	44 ± 5	42 ± 9	0.5 (n.s.)
Triglycerides [mg/dL]	129 ± 23	166 ± 74	0.2 (n.s.)
Fasting glucose [mg/dL]	88 ± 9	101 ± 12	0.02
Fasting insulin [mIU/mL]	9 ± 4	24 ± 7	<0.001
HOMA ^4^	1.3 ± 0.5	3.6 ± 5.3	<0.001

^1^ Body-mass-index (BMI); ^2^ low-density-lipoprotein (LDL); ^3^ high-density-lipoprotein (HDL); ^4^ homeostasis model assessment (HOMA).

**Table 3 ijms-24-06417-t003:** Metabolic parameters of the human cohort used for pyrosequencing and qPCR.

Characteristic	Low-HbA1c (*n* = 65)	High-HbA1c (*n* = 36)	*p* Value
Age [years]	39 ± 12	53 ± 10	<0.0001
Woman [number]	52	21	0.01
BMI ^1^	52.51 ± 11.18	50.54 ± 9.07	0.36 (n.s.)
Total cholesterol [mg/dL]	187.02 ± 32.32	188.31 ± 48.50	0.88 (n.s.)
LDL cholesterol ^2^ [mg/dL]	106.36 ± 30.12	90.67 ± 35.48	0.03
HDL cholesterol ^3^ [mg/dL]	47.02 ± 11.66	41.19 ± 13.45	0.03
Triglycerides [mg/dL]	170.69 ± 89.89	280.81 ± 155.17	<0.0001
Fasting glucose [mg/dL]	94.58 ± 15.99	179.81 ± 71.67	<0.0001
Fasting insulin [mIU/mL]	13.25 ± 11.53	12.61 ± 29.75	0.003
HOMA ^4^	2.9 ± 2.82	12.61 ± 18.82	0.0003

^1^ Body-mass-index (BMI); ^2^ low-density-lipoprotein (LDL); ^3^ high-density-lipoprotein (HDL); ^4^ homeostasis model assessment (HOMA).

## Data Availability

Public data used for Figure 1 are available at https://www.ncbi.nlm.nih.gov/geo (accessed on 30 January 2023) under GSE25401 and GSE20950. Other data are available upon request.

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
