# Peer review of "Fatty Acid Induced Hypermethylation in the Slc2a4 Gene in Visceral Adipose Tissue Is Associated to Insulin-Resistance and Obesity"

_ijms, 2023, doi:10.3390/ijms24076417_

Round 1

Reviewer 1 Report

In this manuscript, Jan, et. al, set out to study the methylation of Slc2a4 in adipocytes. They started with public RNA-seq datasets to screen significant regulated genes between obese and non-obese or IR vs IS group, then further studied the correlation of Slc2a4’s expression level and methylation levels. However, the study is based on a very weak and biased manner, the change of Slc2a4 is less than 2 fold in both obese vs non-obese and IR vs IS, and the mRNA level also decreased less than 2 fold in their own cohort, the difference is significant, but the different is rather minor. These minor changes will lead doubt about the physiological relevance. The in vitro FAs treatment of 3T3-L1 experiment design is confusing, the authors should address why they started treat cells with FAs when induction not after fully differentiated. Above all, the data in this study are insufficient to support the title, there’s no direct evidence to show fatty acid induced hypermethylation of Slc2a4, and there’s no direct evidence to show this methylation promotes insulin resistance. The authors should address the study in an unbiased and logic manner.

Beside above, these major points also need to be aware:

1. The authors omitted the SREBP1c’s function in regulation of de novo lipogenesis in the abstract.

2. Figure 1 a, the schematic failed to illustrate the hypothesis.

3. Langue, especially the abstract needs to be improved.

Minor:

1. Reference the public data. 

Author Response

  1. In this manuscript, Jan, et. al, set out to study the methylation of Slc2a4 in adipocytes. They started with public RNA-seq datasets to screen significant regulated genes between obese and non-obese or IR vs IS group, then further studied the correlation of Slc2a4’s expression level and methylation levels. However, the study is based on a very weak and biased manner, the change of Slc2a4 is less than 2 fold in both obese vs non-obese and IR vs IS, and the mRNA level also decreased less than 2 fold in their own cohort, the difference is significant, but the different is rather minor. These minor changes will lead doubt about the physiological relevance.

Answer: We thank the reviewer for raising concerns about the magnitude of SLC2A4 downregulation. In the field of metabolic diseases including obesity and type 2 diabetes (T2D) it is rather common to find small changes in gene expression that are below 2-fold. This is because obesity and T2D are multifactorial and polygenic diseases. Other authors found similar changes in SLC2A4 expression that ranged from 1.3 to 3-fold and documented clear physiological effects such as altered glucose uptake (https://doi.org/10.1210/jc.2018-02165). Additionally, a 2-fold reduction in Slc2a4 expression in adipocytes was shown to decrease GLUT4 protein levels and glucose uptake, leading to impaired glucose homeostasis (https://doi.org/10.1016/j.mce.2019.05.006). We therefore disagree to consider our 2-fold decrease in SLC1A4 expression as a minor change in gene expression. It also needs to be kept in mind that our human cohort consists of obese subjects with l-HbA1c versus h-HbA1c. Given the already severe underlying state of obesity a difference of 2-fold reduction between the groups is a large effect compared to for example the public data set from GSE20950.

  1. The in vitro FAs treatment of 3T3-L1 experiment design is confusing, the authors should address why they started treat cells with FAs when induction not after fully differentiated.

Answer: We started the FA treatment of 3T3 cells already during the differentiation because it has been shown that fatty acids are also able to induce epigenetic reprogramming during differentiation (https://doi.org/10.2337/db17-0890, https://doi.org/10.1186/s12944-019-1173-6). Since the serum triglyceride levels correlated with SLC2A4 gene expression in VAT of our human cohort we hypothesized that FA may induce the observed changes in DNA methylation. Treating L3L cells with FA and subsequently analyzing DNA methylation was therefore a straight forward mechanistically approach to test our hypothesis.

  1. Above all, the data in this study are insufficient to support the title, there’s no direct evidence to show fatty acid induced hypermethylation of Slc2a4, and there’s no direct evidence to show this methylation promotes insulin resistance. The authors should address the study in an unbiased and logic manner.

Answer: We respectfully disagree that our study is biased because we included public data-sets that were i) not collected by us but others and ii) were not specifically generated to study SLC2A4. Furthermore, we reproduced findings from literature showing that SLC2A4 expression is downregulated in adipose tissue of humans suffering from obesity or insulin resistance. In our study we set out to identify the underlying mechanisms of this downregulation using validated animal and cell-culture studies. We therefore do not feel, that the study was in any way biased or not logical.

Beside above, these major points also need to be aware:

  1. The authors omitted the SREBP1c’s function in regulation of de novo lipogenesis in the abstract.
    Answer: Due to the word limit of 200 words we had to focus on a short and precise description of the manuscript in the abstract. We therefore decided to explain the role of SREBP in regulating Slc2a4 gene expression in the introduction.

  1. Figure 1 a, the schematic failed to illustrate the hypothesis.

Answer: We are sorry if the reviewer did not understand Figure 1a. We modified the figure to add direction of increase or decrease on gene expression and DNA methylation. Additionally, we included an arrow to indicate the chronological occurrence of events.

  1. Langue, especially the abstract needs to be improved.

Answer: We carefully revised the manuscript and paid special attention to the English writing style.

Minor: 

  1. Reference the public data. 

Answer: Both public data sets are referenced in the method section with the reference numbers 30 and 31 (line 320). Additionally, we referenced the public data with their GSE accession number in the results section (line 94 and line 103).

Reviewer 2 Report

Journal: International Journal of Molecular Science

 Title: Fatty acid induced hypermethylation in the Slc2a4 gene in visceral adipose tissue promotes insulin-resistance in obesity

 The authors summarise and describe in their article the effect of obesity and fatty acids on the DNA methylation of the Slc2a4 gene and the effect on the gene transcription of Slc2a4 and several other genes.

 The data are interesting and many experiments and analyses were done, but there are some points of concern and some points, which need improvement or clarifying.

·        Line 18 - 21: If you hypothesise that "obesity-associated dysregulated DNA methylation in Slc2a4 is causative for decreased Slc2a4 expression" why is it not methylated in morbidly obese subject with low HbA1c? This could be a hint, that it is a secondary effect and occurs together with high HbA1c. Please formulate it more precisely.

Line 25 – 27: "treatment of 3T3 pre-adipocytes with palmitate/oleate during differentiation decreased DNA methylation and increased Slc2a4 expression"
But in line 217 – 221: "
onset of decrease at day 5 in Slc2a4 gene expression ........ findings are coherent with the observations ... DNA methylation and Slc2a4 gene expression correlate inversely" you contradict yourself, and figure 3 shows decreased expression and a decreased methylation (even lower than BSA).
Line 243 – 244: "We confirmed our hypothesis that high concentrations of fatty acids increase DNA methylation in adipocytes"
where? Does not fit with statements above.
Figure 3 b,f,g: BSA control, expression of Slc2a4 stays the same (normalised to Hprt) but methylation goes down. Explanation?
Line 305 – 307: "where fatty acids, .... increase enhancer-associated DNA methylation in the Slc2a4 gene to initiate downregulation of gene expression"
Please describe and discuss results better.

·        Line 51 – 52: "whole body Slc2a4 knock out (Slc2a4- /-) mouse models, ... [10]."
Ref 10 is muscle specific GLUT4 KO, not whole body.

·        Line 63 – 65: "we hypothesized that increased DNA methylation ... obese humans with high HbA1c (>6.5%) leads to impaired transcription factor binding and reduced SLC2A4 gene expression."
Again, you suggest that obesity and high HbA1c are responsible for reduced binding and expression, not fatty acids.

·        Line 112: order of the references does not correlate with the text.

·        Line 118 – 122: " SLC2A4 gene expression is significantly decreased... . FASN and MLXIPL gene expression is similarly decreased (Figure 1g), whereas ACACA gene expression is unaltered. Furthermore, FASN, MLXIPL and ACACA expression correlate positively with SLC2A4 gene expression"
How can ACACA gene expression be unaltered but correlate positively with SLC2A4. If these are the experimental data, then the experiments are not very reproducible or reliable or both results are on the border, but then formulate it more carefully please.

·        Figure 1i: CG1   CG1/2   CG4    above DNA sequence, correct names?

·        Line 167 – 169 and 177 – 179: " Slc2a4 gene expression decreases ... reduced to 0.55 week 4 and further decreases to 0.14 week 12 (Figure 2c). ___ Glut4 protein levels tend to be reduced in VAT of HFD mice without reaching statistical significance, likely due to the high variation of Slc2a4 protein expression within the groups (Figure S6a and b)."
If you have such a clear result regarding gene expression, but high variation in protein expression, either the protein measurement is not reliable or the relation between transcription and translation is very weak and then all the gene expression data are not usefully to judge or explain the amount of protein in the cells.
Please give a better explanation.

·        Figure 2a: According to Fig 2a, the diet starts at week 1, which is unlikely. Please improve.

·        Line 208: It would be helpfully, to have some information regarding the mice as body weight, blood glucose (it is not very reader friendly, if we have to look up or download reference 22 to get this information) and regarding the variable results of Glut4 protein it would be nice to see the weight and fasting blood glucose values of individual mice. Not every mouse gains weight at HFD and not every mouse gets diabetic or pre-diabetic.

·        Line 274 – 275: "DNA methylation negatively correlated with SLC2A4 expression in obese subjects receiving metformin"
According to figure S3a and b there is an increase (n.s.) in SLC2A4 and in DNA methylation.
Figure S3c shows a negative correlation, methylation goes down with increasing (-dCT) values. Why are here the –dCT so much higher, from 5 to 10 and not from -10 to -5 as in figure 1h. Such a large difference in the housekeeping gene expression?
In addition, legend to figure S3c and e does not correspond with graphs.

·        Figure S6: analysing just by eye, it is very difficult to understand the results in (b). Blot week 1, the ratio Glut4 to Hsp90 for chow seems clear higher than for hfd, but in (b) hfd is almost twice chow, the same for week 12. Can you explain this?
Bands in figure S6a do not correspond to bands in non-published or original-images files (Hsp90 w2 and Hsp90 w12), please improve.

Author Response

  1. Line 18 - 21: If you hypothesise that "obesity-associated dysregulated DNA methylation in Slc2a4 is causative for decreased Slc2a4 expression" why is it not methylated in morbidly obese subject with low HbA1c? This could be a hint, that it is a secondary effect and occurs together with high HbA1c. Please formulate it more precisely.
    Answer: We very much agree with the reviewer that if obesity drives the observed changes in DNA methylation we would also need to find increased SLC2A4 methylation in the insulin-sensitive obese group. Unfortunately, we are lacking a lean control group and therefore we cannot quantify the difference in SLC2A4 DNA methylation between lean and obese subjects, independently of their HbA1c-status. Due to this lack of lean control group we included the public data set which contains a lean control group. Nevertheless, DNA methylation was not analysed in this study. Thus, it remains currently unknown how DNA methylation differs between lean/healthy and obese. Assuming to our observations that obese subjects with high HbA1c display higher levels of DNA methylation compared to obese subjects with low HbA1c, we would assume that DNA methylation levels in lean/healthy subjects would be even lower than in morbidly obese with low HbA1c levels. As the reviewer correctly pointed out, an additional secondary effect that is more dependent on the insulin-sensitivity might be at play. We therefore modified the wording of our hypothesis.

  1. Line 25 – 27: "treatment of 3T3 pre-adipocytes with palmitate/oleate during differentiation decreased DNA methylation and increased Slc2a4 expression"
    But in line 217 – 221: "onset of decrease at day 5 in Slc2a4 gene expression ........ findings are coherent with the observations ... DNA methylation and Slc2a4 gene expression correlate inversely" you contradict yourself, and figure 3 shows decreased expression and a decreased methylation (even lower than BSA).

Answer: We thank the reviewer for making us aware of possible errors and contradictions. We checked the data again but could not find that average Slc2a4 gene expression in 3T3 cells after PO treatment is lower compared to the BSA control at any time point (please see Fig. 3b). Only DNA methylation of Slc2a4 falls below the BSA control (Fig. 3 f and g). Although, it seems that Slc2a4 expression remains increased relative to the BSA control during the course of the experiment, the differences between BSA control and PO level off. From D10 on, Slc2a4 is not significantly increased anymore.  We described this finding as ‘relative decrease’ which might have been confusing. We revised our wording and hope that the dynamics of Slc2a4 expression after PO treatment are clearer now.

  1. Line 243 – 244: "We confirmed our hypothesis that high concentrations of fatty acids increase DNA methylation in adipocytes"
    where? Does not fit with statements above.

Answer: Again, we apologize for our confusing statement that originated from focussing rather on the delta between BSA and PO than on the absolute values. Due to the effect of PO  on 3T3 cells we concluded that DNA methylation of Slc2a4 is sensitive to fatty acids at the here investigated CpG sites and therefore concluded that similar effects occur in mouse and human adipose tissue. To be more clear we changed our statement into:

“We confirmed our hypothesis that high concentrations of fatty acids dynamically modify Slc2a4 DNA methylation and expression in adipocytes, which could explain the increased DNA methylation seen in obese humans and DIO mice.”

  1. Figure 3 b,f,g: BSA control, expression of Slc2a4 stays the same (normalised to Hprt) but methylation goes down. Explanation?

Answer: The DNA methylation data are absolute data that are not normalized. The Slc2a4 gene expression data however have been normalized to the expression in the BSA control group at each individual treatment day (the BSA Control is set to 1-fold at each day).

When setting the BSA control to 1-fold only at D5 and leaving all other days relative to D5 (please see the figure above, which is also the new supplemental figure S8), it becomes apparent that Slc2a4 expression in both, BSA control and PO treatment increases over time in 3T3 cells. We believe that this absolute Slc2a4 increase in both groups is due to the differentiation process of the cells. Simultaneously,  DNA methylation is decreasing in both groups, which supports our hypothesis, that DNA methylation and gene expression correlate negatively. We added the information that Slc2a4 gene expression increases also in the BSA control with decreasing DNA methylation in line 232– 235 (“It should be noted that the absolute expression of Slc2a4 (normalized to BSA control of D5) increases in both treatment groups over the course of differentiation (Fig. S8). This matches with the decrease in Slc2a4 DNA methylation observed particularly at CpG2 (Fig. 3g).”).

  1. Line 305 – 307: "where fatty acids, .... increase enhancer-associated DNA methylation in the Slc2a4 gene to initiate downregulation of gene expression"
    Please describe and discuss results better.

Answer: We corrected and extended this sentence and it now reads “Collectively, our findings provide a new epigenetic pathomechanism in the regulation of the Slc2a4 gene in obesity in which dietary or endogenously derived FA may increase enhancer-associated DNA methylation in the Slc2a4 gene to partially cause a downregulation of Slc2a4 expression which ultimately reduces glucose uptake and contributes to insulin resistance.“

  1. Line 51 – 52: "whole body Slc2a4 knock out (Slc2a4- /-) mouse models, ... [10]."
    Ref 10 is muscle specific GLUT4 KO, not whole body.

Answer: We apologize for this error. We corrected our statement and the revised version now reads “The impact of SLC2A4 on systemic insulin sensitivity and inter-organ crosstalk was studied in muscle-specific Slc2a4 knock out (Slc2a4- /-) mouse models”.            

  1. Line 63 – 65: "we hypothesized that increased DNA methylation ... obese humans with high HbA1c (>6.5%) leads to impaired transcription factor binding and reduced SLC2A4 gene expression."
    Again, you suggest that obesity and high HbA1c are responsible for reduced binding and expression, not fatty acids.
    Answer: We now realize that we have to be more careful in our statements and that we have to explain our hypothesis in more detail. Due to the fact that gene expression correlated with serum triglyceride levels in humans (Fig. 1j) and DNA methylation is induced by feeding mice with HFD we believe that FA are a key driver. We revised the sentence accordingly.

  1. Line 112: order of the references does not correlate with the text.

Answer: We are extremely grateful for making us aware of this error. The references are cited at the correct section of the text however the numbers are not chronological throughout the text. This error occured during copying the manuscript in the MDPI template. We revised the numbering and the references are now in chronological order.

  1. Line 118 – 122: " SLC2A4 gene expression is significantly decreased... . FASN and MLXIPL gene expression is similarly decreased (Figure 1g), whereas ACACA gene expression is unaltered. Furthermore, FASN, MLXIPL and ACACA expression correlate positively with SLC2A4 gene expression"
    How can ACACA gene expression be unaltered but correlate positively with SLC2A4. If these are the experimental data, then the experiments are not very reproducible or reliable or both results are on the border, but then formulate it more carefully please.

Answer: ACACA gene expression is not altered when groups are stratified in l-HbA1c and h-HbA1c. However, as often in physiology, gene expression is likely influenced by continuous factors. Thus ACACA expression is associated with HbA1c as clearly shown by the significant correlation.   Nevertheless, we changed the wording to “FASN and MLXIPL gene expression is similarly decreased (Figure 1g), whereas ACACA gene expression is non-significantly decreased if compared between both subgroups in total. Furthermore, FASN and MLXIPL expression correlate positively with SLC2A4 gene expression (Figure 1h, SLC2A4/FASN: R = 0.3958, p = 0.0004; SLC2A4/MLXIPL: R = 0.3261). Though the group-wise difference of ACACA expression was not significant, we observed a weak significant positive correlation due to the reduced scattering (SLC2A4/ACACA: R = 0.3818, p = 0.0002).”

This should emphasise that even though ACACA gene expression is unaltered between the subgroups, the connection between ACACA and SLC2A4 gene expression is still present. In human subjects, higher levels of SLC2A4 gene expression are accompanied by higher levels of ACACA gene expression.

  1. Figure 1i: CG1 CG1/2 CG4    above DNA sequence, correct names?

Answer: We apologize for this mistake. The label should be CG1   CG2/3     CG4. We corrected the error.              

  1. Line 167 – 169 and 177 – 179: " Slc2a4 gene expression decreases ... reduced to 0.55 week 4 and further decreases to 0.14 week 12 (Figure 2c). ___ Glut4 protein levels tend to be reduced in VAT of HFD mice without reaching statistical significance, likely due to the high variation of Slc2a4 protein expression within the groups (Figure S6a and b)."
    If you have such a clear result regarding gene expression, but high variation in protein expression, either the protein measurement is not reliable or the relation between transcription and translation is very weak and then all the gene expression data are not usefully to judge or explain the amount of protein in the cells.
    Please give a better explanation.   
    Answer: As the reviewer suggested correctly, the protein measurement was not reliable, likely due to fat contamination, which was visible as fat drops on the top of the protein lysate. We tried extra steps of centrifugation to reduce the fat contamination. Moreover, we tested different methods for protein quantification, including the BCA method (ThermoFisher, Pierce™ BCA Protein Assay Kit, 23227) and methods relying on fluorescence (QIAGEN, Qubit™ Protein and Protein Broad Range, Q33211). Nevertheless, we could not get reliable quantification of protein, which can be clearly seen in the varying HSP90 levels. Therefore, we decided to show the data only as supplemental figure and not in the main manuscript. With the mentioned description in the manuscript (…,likely due to the high variation of Slc2a4 protein expression within the groups’), we intended to interpret the WesternBlot data with caution. We now realize that we should rather mention the difficulties in protein quantification and changed the wording accordingly.

  1. Figure 2a: According to Fig 2a, the diet starts at week 1, which is unlikely. Please improve.

Answer: The diet started at week 0 and animals at time point 1 were exposed to HFD for exactly one week. We reworked the figure and added the time point -1 week to indicate the acclimatisation phase of the animals. Moreover, we added a 0 week time point indicating the start of the diet intervention to address this issue. New figure 2a:

  1. Line 208: It would be helpfully, to have some information regarding the mice as body weight, blood glucose (it is not very reader friendly, if we have to look up or download reference 22 to get this information) and regarding the variable results of Glut4 protein it would be nice to see the weight and fasting blood glucose values of individual mice. Not every mouse gains weight at HFD and not every mouse gets diabetic or pre-diabetic.

Answer: We absolutely agree that it is not reader friendly to omit the metabolic phenotyping of the mice and consequently added the data to the current manuscript. We now provide the body weight, plasma insulin, hepatic triglyceride levels and GTT in a separate supplementary figure FS7. Throughout the high-fat feeding, all mice persistently gained weight and plasma insulin levels increased from week 8 on. Nevertheless, this increase in plasma insulin was heterogeneous because mice were not fasted but fed ad libitum at the time of blood sampling. Thus, we decided to indicate different metabolic stages with a coloured background in the figure F3a with grey being a phase of initial challenge, green being compensation and purple being failure to compensate.           

  1. Line 274 – 275: "DNA methylation negatively correlated with SLC2A4 expression in obese subjects receiving metformin"
    According to figure S3a and b there is an increase (n.s.) in SLC2A4 and in DNA methylation.
    Figure S3c shows a negative correlation, methylation goes down with increasing (-dCT) values. Why are here the –dCT so much higher, from 5 to 10 and not from -10 to -5 as in figure 1h. Such a large difference in the housekeeping gene expression?
    In addition, legend to figure S3c and e does not correspond with graphs.

Answer: Thanks for making us aware of inconsistencies. After checking the data again, we noticed that the x-axis in S3c was wrongly labelled. The correlation is not with -dCt, but with the dCt values. To make it uniformly, we changed the correlation in 1h also to the dCt values.  
We also corrected the figure legend for 3c and e.      

  1. Figure S6: analysing just by eye, it is very difficult to understand the results in (b). Blot week 1, the ratio Glut4 to Hsp90 for chow seems clear higher than for hfd, but in (b) hfd is almost twice chow, the same for week 12. Can you explain this?

Bands in figure S6a do not correspond to bands in non-published or original-images files (Hsp90 w2 and Hsp90 w12), please improve.

Answer: Throughout the Western Blot analysis, we were facing problems with reliable protein quantification as described in the answer above. Within each group, the variability of loaded protein was very high. For week 1, three HDF-mice had enormously increased Glut4/Hsp90 ratios, whereas the rest of the HFD group displayed very low Glut4/Hsp90 ratios, when compared to the control chow diet. Those three high-fat fed mice had increased Glut4/Hsp90-signal ratio between 30 to 50 indicating increased mean Glut4 protein levels. In the figure, the mean ± SEM was plotted, which does not exactly display the high variance, when compared with the standard deviation. Similar findings were observed for week 12. It is also noteworthy that only four representative samples per group are shown in the membrane pictures (FS6a). In total eight animals per group were analysed (FS6b). Unfortunately, the animals, which deviated from the mean, were not identified as outliers, due to the underlying high variance. For analysing the data we used a highly sensitive software (Image Lab), which also has a function for background subtraction. Accordingly, bands which may seem very intense to the naked eye may have higher background signals, which is subtracted from the band-intensity. We thus emphasize to interpret the Western Blot data with caution, due to the high variation as described in the manuscript.

We compared again the labelling of the Western Blot images shown in Figure S6 and the original Western Blots. Accidentally, the housekeeping protein Hsp90 from week 2 and week 12 are swapped. We corrected this mistake. Nevertheless, for the analysis of the data the correct housekeeping protein images were used. As additional information, we added the molecular weight to improve readability of the figure.

 Reworked:

Reviewer 3 Report

In this study, authors investigated connection of the hypermethylation in the Slc2a4 gene in visceral adipose tissue and insulin-resistance in obesity. Study is well conducted, however, several issues should be addressed:  

In the Introduction, every information that describes the conduction of the study, or the Results of the study should be moved to more appropriate part of the paper (e. g. Discussion)

  In the Results, text is too broadly written - most important information should be highlighted better   In the Discussion, better emphasis should be put into clinical implications that could be derived from these analyses.    Moreover, limitations of this study are not pointed out enough. Please revise.

Author Response

  1. In the Introduction, every information that describes the conduction of the study, or the Results of the study should be moved to more appropriate part of the paper (e. g. Discussion)
    Answer: The partial description of the results in the introduction are required by the MDPI criteria (MDPI criteria for introduction: ,Finally, briefly mention the main aim of the work and highlight the principal conclusions’). We therefore keep the introduction unaltered.

  1. In the Results, text is too broadly written - most important information should be highlighted better.   In the Discussion, better emphasis should be put into clinical implications that could be derived from these analyses.    Moreover, limitations of this study are not pointed out enough. Please revise.

Answer: We agree that within the presented data, the main finding of each conducted experiment may get lost. Therefore we included a final paragraph in each result part summarizing the most important findings to ease readability. These new conclusions are:

 “ 2.1.: ,In conclusion, SLC2A4 gene expression is reduced in the VAT of obese individuals with high HbA1c and is accompanied by a hypermethylation at a SP1/SREBF binding site. Hence SLC2A4 gene expression correlates with serum triglyceride levels, SLC2A4 might be regulated by fatty acid induced DNA methylation.”

“2.2.: , Overall, the findings observed in human are reproducible in our longitudinal mouse model of DIO, where increased DNA methylation in the ROI is associated with a lower expression of the Slc2a4 gene. Further, in the mouse model there is a clear correlation between DNA methylation and gene expression, pointing at an epigenetically regulation of the Slc2a4 gene that may be masked in the human cohort, due to different medications or lifestyle factors. Further, the mouse model shows that these epigenetic dysregulation establishes in a very early stage of obesity prior to the onset of insulin resistance. FFA might regulate SLC2A4 gene expression via altering DNA methylation in the ROI since Slc2a4 gene expression and DNA methylation correlate with hepatic triglyceride content in mice and with free fatty acids (FFA) plasma levels in humans.”

“2.3.:, In the fully methylated state the luciferase signal decreases to 77.53% (± 16.71%, p = 0.039). This result aligns with the previous findings in mouse, humans and differentiating 3T3 preadipocytes, where higher levels of DNA methylation are associated with lower Slc2a4 gene expression.’). “

To better emphasis the clinical implications, we added a final paragraph to the discussion:           
“Our findings may also have implications for dietary therapies in diabetes, especially regarding the diet composition. Often, the emphasis is on the reduction of carbohydrate intake, since the primary symptom in T2D is a dysregulated glucose homeostasis.”

We also added a further paragraph to highlight the limitations of the study better:          
“A further limitation of this study is the use of 3T3-derived adipocytes in the in vitro experiments. It still has to be elucidated the effect of FA in Slc2a4-DNAme in maybe primary adipocytes of obese subjects. (Line 308-310).”

Reviewer 4 Report

In the manuscript entitled “Fatty acid induced hypermethylation in the Slc2a4 gene in visceral adipose tissue promotes insulin-resistance in obesity” Britsemmer and coworkers took an effort to determine the effect of high fat diet )mainly consisting of palmitate/oleate)  on epigenetic mechanism in the regulation of the Slc2a4 gene in obesity. The manuscript contains very interesting and important data and is potentially noteworthy for the readers, especially since it matches data collected from transcriptome of human subcutaneous adipose tissue (available and collected during in vivo study), in vivo animal based study (mice), and in vitro cell line based study (3T3-adiopcytes). The manuscript present results obtained from tremendous work done by co-authors, and it should be emphasized – congratulations! It is well written and easy to follow, however it requires some modifications before being accepted for publication. If possible, the answers and comments should be included in a new version of manuscript, since they will allow simpler following of presented data.

First, please add the supplementary data directly to the manuscript – already there are many results presented, but addition of supplementary data will make all taken steps clearer for the reader.  Please, in the manuscript comment the results obtained on the protein level with western blot.

Have the Authors performed the experiments (even with 3T3L1 adipocytes) in the presence of palmitate/oleate with silenced/overexpressed Slc2a4 gene? Or in the presence of specific inhibitors? Have authors stained differentiated adipocytes and measured glucose uptake? Please, add the results.

All figures obligatory need to be enlarged! In Figure S6a add mw of each protein and explain “c” and “h”.

Please check the explanation of used short forms of, i.e. enzymes, genes presented for the first time.  

In the Methods describe the details of dietary pattern used for mice (HFT).

Please, present the Figure captures according to the journal template.

In summary, the manuscript requires the major revision. 

Author Response

  1. First, please add the supplementary data directly to the manuscript – already there are many results presented, but addition of supplementary data will make all taken steps clearer for the reader. Please, in the manuscript comment the results obtained on the protein level with western blot.
    Answer: We thank the reviewer for the kind words and for encouraging us to move the supplemental data to the main manuscript. Unfortunately, we are facing space limits and thus we need to keep the supplemental data as supplement. To enhance readability we split the first figure up into two figures and also enlarged the remaining figures.            
    We commented the western blot results in line 182-184 as followed: , Glut4 protein levels tend to be reduced in VAT of HFD mice without reaching statistical significance, likely due to fat contamination of the protein lysate which influenced the protein quantification and led to unequal amounts of protein loaded to the gel (indicated by varying Hsp90 band intensities, Figure S6a and b).’
  2. Have the Authors performed the experiments (even with 3T3L1 adipocytes) in the presence of palmitate/oleate with silenced/overexpressed Slc2a4 gene? Or in the presence of specific inhibitors? Have authors stained differentiated adipocytes and measured glucose uptake? Please, add the results.

Answer: Those are some very interesting experimental suggestions. However, we decided to not consistently overexpress Slc2a4 in vitro or in vivo, since the effects of selective Slc2a4 overexpression in adipose tissue or inducing Slc2a4 expression in 3T3 has already been described (doi: 10.1152/ajpendo.00116.2005, DOI: 10.1530/JME-17-0041). Also silencing Slc2a4 in 3T3 adipocytes with a shRNA lead to reduced de novo lipogenesis (https://doi.org/10.1210/en.2005-1638). For our studies, excessively increased Glut4 achieved by Slc2a4 overexpression would uncouple the obtained results from any epigenetic mechanism. We did not measure glucose uptake in 3T3 cells.

  1. All figures obligatory need to be enlarged! In Figure S6a add mw of each protein and explain “c” and “h”.
    Answer: We agree that the figures are too small and hard to read. We decided to divide Figure 1 into two separated Figures, one for the public available transcriptomic data and the other for the analysis of our human cohort. Additionally, we enlarged all other figures. We added the MW to figure S6 and explained in the caption ‘c’ = Chow and ‘h’ = high fat diet.

  1. Please check the explanation of used short forms of, i.e. enzymes, genes presented for the first time.
    Answer: We added the full name for each gene and protein when first mentioned and introduced the abbreviation in brackets. After this, we stick to the abbreviation.

The revised version now reads:,The solute carrier family 2 member 4 (SLC2A4) gene, encoding for the insulin-dependent glucose transporter type 4 (GLUT4) in muscle…’

  1. In the Methods describe the details of dietary pattern used for mice (HFT).

Answer: We described the dietary pattern in the methods as followed in line 354-358: ,Mice had ad libitum access to water and food. After one week of acclimation, mice were randomized into two groups of similar body weights. Mice were fed ad libitum with either chow diet (= control group; Breeding Diet 1314 obtained from Altromin, Germany) or 60 % high-fat diet (= treated group; HFD, D12492, Research Diets, USA) for 1, 2, 4, 5, 6, 7, 8 or 12 weeks.  To clarify that a mice was persistently fed under one of the diets, we added the information ,Mice were consistently fed ad libitum with either chow or 60% high-fat diet…’

We now also include metabolic phenotypes of the mice in Figure S7.

  1. Please, present the Figure captures according to the journal template.

Answer: We corrected the captions according to the journal template with e.g. (a), (b), (c)
Additionally, we used ‘;’ as a separator between figure captions, instead of ‘.’ This is indicated in the MDPI template. E.g.: (a) … ; (b) … ; (c) … 

Round 2

Reviewer 1 Report

In the revised manuscript, the authors made substantial improvement, however, the title is still not supported by the present data.

This manuscript is a correlation study, therefore, the causal relationship between methylation of Slc2a4 gene and insulin resistance is overstatement.

Major:

1. Fig 3 mouse study, the normalization is not proper, with current presentation, we could not tell the difference is caused by decrease transcription of related genes in HFD group or increase of the genes in Chow group. All the points should be normalized to week 1 Chow group.

2. Fig 3 mouse data better show the insulin sensitivity of the mice, which may support the methylation and decrease of Slc2a4 occurs before insulin resistance.

3.  Fig 4 3T3 study, treating P/O from the beginning creates a confounding variant, the differentiation of the pre-adipocytes, the authors should prove there’s no effects on differentiation by treating P/O. High concentration of fatty acids causes lipotoxicity, authors should also show the difference is not caused by this. Show the WB of Glut4.

4. Language still need to be improved, try to avoid long complicated sentences, especially in abstract and result section.

Minor:

1. Authors should address how they prepare P/O in the method.

Author Response

Comment: In the revised manuscript, the authors made substantial improvement, however, the title is still not supported by the present data.

This manuscript is a correlation study, therefore, the causal relationship between methylation of Slc2a4 gene and insulin resistance is overstatement.

Answer: We agree with the reviewer that the human data are only of correlative nature. However, the longitudinal mouse study shows that DNA methylation significantly increases prior to the rise of plasma insulin levels, establishment of insulin resistance and decrease of Slc2a4 gene expression under HFD. Moreover, the physiological role of adipose tissue Slc2a4 expression on systemic insulin sensitivity is well established by numerous publications, which we describe in detail in the introduction. Nevertheless, we edited the title and it now reads: “Fatty acid induced hypermethylation in the Slc2a4 gene in visceral adipose tissue is associated to insulin-resistance and obesity

Major:

Comment: Fig 3 mouse study, the normalization is not proper, with current presentation, we could not tell the difference is caused by decrease transcription of related genes in HFD group or increase of the genes in Chow group. All the points should be normalized to week 1 Chow group.

Answer: We thank the reviewer for suggesting normalizing to week 1 chow in the DIO mouse model. However, we explicitly decided to normalize the data to each week chow for the following reasons. The visceral adipose tissue is a highly plastic tissue, which undergoes remodeling during ageing (https://doi.org/10.1016/j.cell.2021.12.016, DOI: 10.1113/jphysiol.2014.274175). This makes week 1 chow an inappropriate reference group for all subsequent time points, because e.g. mice with 16 weeks of age (12 week group) would be normalized to 4 week old mice (week 1 group). Additionally, this would distort the dietary effect of HFD, which we are particularly interested in. By normalizing the HFD mice to their age matched chow control we exclude any effects of aging. Nevertheless, to address any concerns about the normalization, we now provide normalization of Slc2a4 gene expression to chow week 1 in the supplementary figure S5a (please see also the figure below). Importantly, the decrease of Slc2a4 expression in the HFD group is clearly visible (especially at the later time points) even when the entire data is normalized to chow week 1. We decided to leave the main figure as it is, due to the above mentioned points. We also want to note that the correlation of gene expression with different parameters (DNA methylation, glucose and insulin levels, etc.) were performed using the -dCt values. This has the benefit that correlation analysis is completely independent to the chosen normalization. Therefore, our original findings and results that high levels of DNA methylation correlate with low levels of gene expression remain valid.

Comment: Fig 3 mouse data better show the insulin sensitivity of the mice, which may support the methylation and decrease of Slc2a4 occurs before insulin resistance.

Answer: We agree that only providing the GTT and plasma insulin levels in figure S8 may not be enough to document states of insulin resistance. Therefore, we calculated the Homeostasis Model Assessment of insulin resistance (HOMA-IR) index of all mice as described in doi:10.3164/jcbn.09-83. The HOMA-IR data are now shown in the main manuscript in figure 3b (please see also below).

Comment:  Fig 4 3T3 study, treating P/O from the beginning creates a confounding variant, the differentiation of the pre-adipocytes, the authors should prove there’s no effects on differentiation by treating P/O. High concentration of fatty acids causes lipotoxicity, authors should also show the difference is not caused by this. Show the WB of Glut4.

Answer: We appreciate the comment to consider effects of lipotoxicity or differentiation on the fatty acid induced DNA methylation in 3T3. As the reviewer points out correctly, we cannot determine the exact molecular mechanism by which fatty acids induce DNA methylation, as we stated in our manuscript: “Besides the fact that DNA methylation is changing during differentiation [29], this process seems to be accelerated by PO treatment in 3T3 preadipocytes” (Line 312 – 315), and “Chronic conditions of high fatty acid levels on the other hand, which frequently occur in obesity and are termed lipotoxicity, lead to a decrease in Slc2a4 gene expression in the mouse and cell culture model” (Line 319 – 322). However, both, lipotoxicity and differentiation of progenitor adipocytes, are common processes altered in obesity and type 2 diabetes and are likely caused by an excessive abundance of fatty acids (doi:10.3390/ijms20092358).  Lipid-overload also leads to reduced Glut4-mediated glucose uptake in 3T3 cells (https://doi.org/10.1128/MCB.01321-14).

Since the role of Slc2a4 and Glut4-mediated glucose uptake in FA-challenged 3T3 cells has widely been studied, we purely focused on the DNA methylation and gene expression aspect to also give a possible explanation for the observed changes in DNA methylation and gene expression observed in humans and mice. Regardless of the exact mode of action, we found that fatty acids dynamically decrease DNA methylation in the Slc2a4 gene. Throughout the treatment period, those differences normalize again. The regulatory function of DNA methylation on gene expression is finally proven with our methylation sensitive reporter gene assay as well. We consider those results for highly important in the scientific community.           
Because we cannot pinpoint the exact molecular mechanism by which fatty acids act on DNA methylation we added an additional sentence in the discussion about further limitations in our in vitro experiments:    

Line 326 – 327: “Additionally, we cannot determine the exact mechanism on how DNA methylation is dynamically changed by fatty acids.”

Comment: Language still need to be improved, try to avoid long complicated sentences, especially in abstract and result section.

Answer: We revised the language again and paid attention to long sentences.

Minor:

Comment: Authors should address how they prepare P/O in the method.

Answer: We added information about the P/O preparation to the method part “4.8 Treatment of 3T3 preadipocytes with palmitate/oleate mixture.

Line 489 – 490: Palmitate (Sigma, P9767) and oleate (Sigma, O7501) were solved in 0.1 M NaOH at 70 °C and diluted in 7.5 % BSA to final concentration of 5 mM.        

Reviewer 3 Report

The MS is improved in several aspects, but there are still several issues which must be resolved:

As I already stated in my previous report (what the authors refused to change in R1) is correction of the Introduction section.

Additionally, in Figure 3 mouse study, the authors must correct normalization, because it is not clear if the difference is caused by decreased transcription of related genes in the HFD group or increase of the genes in Chow group.

Author Response

Comment: As I already stated in my previous report (what the authors refused to change in R1) is correction of the Introduction section.

Answer: We understand the point of criticism from the previous round or revision and we surely did not ignore this comment. However, we did not change the introduction in the first round of revision because it is mandatory by MDPI guidelines to include a description of the aims, results and conclusion already in the introduction. To find a compromise between journal requirements and the reviewer we shortened the description of the results in the introduction.

It now reads: “Accordingly, we hypothesized that DNA methylation of the SLC2A4 in the proximity of a SP1 and SREBP-1 binding motif is increased in VAT of obese humans. This might lead to impaired transcription factor binding and reduced SLC2A4 gene expression. Indeed, we found that DNA methylation is dynamically altered and associated to reciprocal Slc2a4 expression and fatty acids in obese humans and mice with insulin resistance.
Herewith, we describe a novel epigenetic mechanism regulating the SLC2A4 gene by fatty acid induced alterations in DNA methylation, which may play a role in the development of insulin resistances in obesity.” (Line 64 – 85)

Comment: Additionally, in Figure 3 mouse study, the authors must correct normalization, because it is not clear if the difference is caused by decreased transcription of related genes in the HFD group or increase of the genes in Chow group.

Answer: We thank the reviewer for suggesting normalizing to week 1 chow in the DIO mouse model. The normalization also raised concerns by reviewer #1.   
However, we explicitly decided to normalize the data to each week chow for the following reasons. The visceral adipose tissue is a highly plastic tissue, which undergoes remodeling during ageing (https://doi.org/10.1016/j.cell.2021.12.016, DOI: 10.1113/jphysiol.2014.274175).

This makes week 1 chow an inappropriate reference group for all subsequent time points, because e.g. mice with 16 weeks of age (12 week group) would be normalized to 4 week old mice (week 1 group). Additionally, this would distort the dietary effect of HFD, which we are particularly interested in. By normalizing the HFD mice to their age matched chow control we exclude any effects of aging. Nevertheless, to address any concerns about the normalization, we now provide normalization of Slc2a4 gene expression to chow week 1 in the supplementary figure S5a (please see also the figure below). Importantly, the decrease of Slc2a4 expression in the HFD group is clearly visible (especially at the later time points) even when the entire data is normalized to chow week 1. We decided to leave the main figure as it is, due to the above mentioned points. We also want to note that the correlation of gene expression with different parameters (DNA methylation, glucose and insulin levels, etc.) were performed using the -dCt values. This has the benefit that correlation analysis is completely independent to the chosen normalization. Therefore, our original findings and results that high levels of DNA methylation correlate with low levels of gene expression remain valid.

Reviewer 4 Report

I have read the Authors’ response and the new version of manuscript. The Authors answered most of my concerns, but the Figures presented are still too small or sometimes not readable (Figure 4). I would suggest to perform additional in vitro experiment with 3T3-L1 cell line, which will confirm the effect of palmitate/oleate on glucose uptake using, i.e. fluorescent NBDG probe or other glucose-based assay. The corresponding experiment can be done also with selected by Authors DNA methylation inhibitor, and after addition of insulin to culture medium to induce insulin-resistance. It will greatly enrich the already performed tremendous work of Kirchner and coworkers, and demonstrate the functional effect of FA on the glucose uptake, not only on Slc2a4 level in adipocytes. Finally, could you please add information about the source of 3T3-L1 cells used in the study?

In summary, I suggest the major revision of the manuscript.

Author Response

Comment: I have read the Authors’ response and the new version of manuscript. The Authors answered most of my concerns, but the Figures presented are still too small or sometimes not readable (Figure 4).

Answer: We apologize that the changes (enlarging the figure sizes) did not increase the readability. We now also increased the font size of each individual graph in figure 3 and 4 and hope that this improved readability.

Comment: I would suggest to perform additional in vitro experiment with 3T3-L1 cell line, which will confirm the effect of palmitate/oleate on glucose uptake using, i.e. fluorescent NBDG probe or other glucose-based assay. The corresponding experiment can be done also with selected by Authors DNA methylation inhibitor, and after addition of insulin to culture medium to induce insulin-resistance. It will greatly enrich the already performed tremendous work of Kirchner and coworkers, and demonstrate the functional effect of FA on the glucose uptake, not only on Slc2a4 level in adipocytes.

Answer: We appreciate the suggestion for further experiments in 3T3 cells. For the in vitro experiments in 3T3 cells, we focused on the connection between DNA methylation and Slc2a4 gene expression and how DNA methylation is dynamically changed by treatment with fatty acids to provide a possible explanation for the observed changes in DNA methylation and gene expression in human and mice. As Glut4-dependent glucose uptake in 3T3 cells after FA treatment has already been investigated, we purely focused on the DNA methylation aspect (doi.10.1055/s-2007-978793; doi.10.1210/endo.138.10.5458; doi.10.1128/MCB.01321-14).
The use of DNA methylation inhibitors (DNMT-inhibitors) is a very interesting thought. In previous projects, we tested the treatment of 3T3 cells with 5-azacytidine (5-azaC) which is a known inhibitor of the DNA methyltransferase 1 (DNMT1) causing hypomethylation (doi.10.3324/haematol.2012.074823). Unfortunately, we did not find significant effects on DNA methylation in 3T3 cells, since 5-azaC acts only in proliferating cells (G1-phase), and once the differentiation in 3T3 cells is induced, 3T3 cells remain in the G0-phase for differentiation. Most published studies using 5-azaC are thus conducted in proliferating cells. Additionally, one has to keep in mind that DNA methylation inhibitors cause global hypomethylation since there is no locus specific mode of action. This introduces a variety of off-target effects, which would make it even more difficult to trace back the effect on enhanced or repressed glucose uptake in 3T3 cells on Slc2a4 DNA methylation. Therefore, we decided to not perform further in vitro experiment.

Comment: Finally, could you please add information about the source of 3T3-L1 cells used in the study?

Answer: We apologize that we forgot to mention the origin of the 3T3-L1 cells in our manuscript. We added this information in the method section: Line 528: , 3T3-L1 preadipocytes (ATCC, Germany)  were cultured in proliferation medium (gibco DMEM with 4.5g/l glucose, 10 % FBS, 1 % Pen/Strep) at 37°C and 5% CO2’.

Round 3

Reviewer 1 Report

In this revision, the authors tried to answer most of the questions, however, it revealed the inconsistency of data in Fig 3, and the authors refused to do experiments to get rid of confounding variants in Figure 4.

Major:

1. Fig 3 and Fig S5, different normalization methods can have totally different conclusions, if  “high levels of DNA methylation correlate with low levels of gene expression” is true, the methylation level of Slc2a4 in chow diet group should decrease, but in Fig 3 i and j, there’s no decrease. The normalization method author chosen to use will mislead readers.

2. Fig 4, the literatures referred by the authors treated FFAs to differentiated 3T3-L1 cells, not during the differentiation. To determine if there’s effects on differentiation, the authors could detect differentiation markers either by WB or qPCR, it’s not difficult to perform those experiments. Besides, HFD induced obesity experiments mostly are done in adult mice (including you), under which condition, the differentiation of adipocytes are mostly finished. Therefore, it’s more reasonable to do this experiment in differentiated 3T3-L1. 

Follows the authors’ logic,  HFD increased methylation of Slc2a4 and decreased its transcription and expression in mice, and tried to use fatty acids treatment to mimic this process in 3T3-L1 cells, but why they showed different results, why PO treated 3T3-L1 showed decreased methylation and increased transcription of Slc2a4. Fig 4 does not benefit the manuscript, and it was not designed properly.

Reviewer 3 Report

No further comments.

Reviewer 4 Report

The Authors answered my concerns and I accept their explanations hoping to find their further studies in the area of DNA methylation regulation.